# Benchmarking and Analyzing 3D Human Pose and Shape Estimation Beyond Algorithms

**Hui En Pang**[1,2], **Zhongang Cai**[1,2], **Lei Yang**[2], **Tianwei Zhang**[1], **Ziwei Liu**[1 ✉]

[1]S-Lab, Nanyang Technological University    [2]SenseTime Research

{huien001, tianwei.zhang, ziwei.liu}@ntu.edu.sg
{caizhongang, yanglei}@sensetime.com

## Abstract

3D human pose and shape estimation (a.k.a. "human mesh recovery") has achieved substantial progress. Researchers mainly focus on the development of novel algorithms, while less attention has been paid to other critical factors involved. This could lead to less optimal baselines, hindering the fair and faithful evaluations of newly designed methodologies. To address this problem, this work presents the *first* comprehensive benchmarking study from three under-explored perspectives beyond algorithms. *1) Datasets.* An analysis on 31 datasets reveals the distinct impacts of data samples: datasets featuring critical attributes (*i.e.* diverse poses, shapes, camera characteristics, backbone features) are more effective. Strategical selection and combination of high-quality datasets can yield a significant boost to the model performance. *2) Backbones.* Experiments with 10 backbones, ranging from CNNs to transformers, show the knowledge learnt from a proximity task is readily transferable to human mesh recovery. *3) Training strategies.* Proper augmentation techniques and loss designs are crucial. With the above findings, we achieve a PA-MPJPE of 47.3 $mm$ on the 3DPW test set with a relatively simple model. More importantly, we provide strong baselines for fair comparisons of algorithms, and recommendations for building effective training configurations in the future. Codebase is available at `https://github.com/smplbody/hmr-benchmarks`.

## 1 Introduction

3D human pose and shape estimation (a.k.a. "human mesh recovery"[1]) has attracted a lot of interest due to its vast applications in robotics, computer graphics, AR/VR, etc. Common approaches take monocular RGB images [33, 30, 35, 59] or videos [32, 31, 48] as input to regress the parameters of a human body model. One of the most popular human parametric models is SMPL [47]. Over the years, a substantial amount of novel algorithms have been proposed [36, 18, 35, 24, 8, 54, 29, 37, 12, 34, 33], which significantly improve the recovery accuracy.

Despite the advances in mesh recovery algorithms, prior works rarely systematically investigated other fundamental factors that are also crucial to the model performance. (1) Different selections of datasets and their contributions yield distinct model performance. This is especially prominent in human mesh recovery as datasets containing different label modalities (2D keypoints, 3D keypoints, mask, SMPL parameters) are usually combined for training. (2) The mesh recovery model is commonly learnt from a pretrained backbone. The quality of the backbone (e.g., network architecture, weight initialization) is a primary determinant of the downstream task. (3) The performance of the mesh recovery model is also highly sensitive to the training strategies, including data augmentation and training loss design. *It is still unclear how these factors can affect the model performance and what are the optimal training configurations to obtain good mesh recovery models.*

---

[1]The two terms are used interchangeably in this work.

36th Conference on Neural Information Processing Systems (NeurIPS 2022) Track on Datasets and Benchmarks.

Such a lack of understanding can severely impede the development of mesh recovery research. First, researchers may build and assess new algorithms with less optimal training configurations, which cannot fully reflect the benefits of the new inventions. For instance, the state-of-the-art algorithms SPIN [35] and PARE [33] can achieve the PA-MPJPE (*i.e.*, recovery error) of 59.2 $mm$ and 50.9 $mm$, respectively, while we can obtain the PA-MPJPE of 47.3 $mm$ by selecting a better configuration with a simple base method (Table 1). Second, some prior works compare different algorithms or methods with different training configurations, leading to unfair evaluations. For instance, HMR [30] and SPIN [35] are often used as the baselines for comparison with various algorithms [33, 34, 29, 40, 12] despite having used vastly different dataset mixes. There are fewer studies [83, 34] that utilize the same dataset mix as HMR or SPIN or replicate their dataset mix with HMR for ablation.

To address the aforementioned problems, we perform a large-scale benchmarking study about human mesh recovery tasks from three perspectives. **(1) Datasets.** We provide comprehensive evaluations on 31 datasets, including several that have not been used for mesh recovery. We observe that huge performance gains can be achieved from a careful selection of datasets. We identify factors that make a dataset competitive, and provide suggestions to enhance existing datasets or collect new ones. **(2) Backbone.** Mainstream approaches are still using conventional CNN-based feature extractors [33, 12]. We extend the study to 10 backbone architectures, including vision transformers. We also investigate the effect of pretraining and discover that weight initialization from a strong pose estimation model is highly complementary for mesh recovery tasks. **(3) Training strategy.** We examine different augmentations and training loss designs. We discover that L1 loss is more effective for supervision and curbing noise than the typically used mixed losses. We explain the effectiveness of different augmentations based on the underlying feature distributions of the train and test datasets.

Putting together our findings, we establish strong baselines for different dataset mixes and backbones on the HMR algorithm [30] and 3DPW test set [75], as shown in Table 1 (results on the H36M test set [23] can be found in the appendix). Patel et al. [58] suggested that 3DPW-test benchmarks

Table 1: **Our identified optimal baseline models with the performance on the 3DPW test set.** Abbreviations for the datasets - Human3.6M [23]: H36M, MPI-INF-3DHP [51]: MI, MuCo-3DHP [52]: MuCo, PoseTrack [2]: PT, OCHuman [86]: OCH

| Algorithm | Dataset | Backbone | PA-MPJPE↓ | MPJPE↓ | PA-PVE↓ | PVE↓ |
|---|---|---|---|---|---|---|
| PARE [33] | EFT-[COCO, LSPET, MPII], H36M, SPIN-MI | HrNet-W32 | 50.90 | 82.0 | - | 97.9 |
| Ours | EFT-[COCO, LSPET, MPII], H36M-Aug, SPIN-MI | HrNet-W32 | 47.68 | 81.16 | 64.70 | 98.23 |
| SPIN [35] | H36M, MI, COCO, LSP, LSPET, MPII | ResNet-50 | 59.2 | 96.9 | - | 135.1 |
| HMR [30] | H36M, MI, COCO, LSP, LSPET, MPII | ResNet-50 | 76.7 | 130.0 | - | - |
| Ours | H36M, MI, COCO, LSP, LSPET, MPII | ResNet-50 | 51.66 | 82.80 | 70.53 | 100.59 |
| Ours | H36M, MI, COCO, LSP, LSPET, MPII | Twin-SVT-B | 48.77 | 82.91 | 66.91 | 96.33 |
| Ours | H36M, MI, COCO, LSP, LSPET, MPII | HrNet-W32 | 49.18 | 79.76 | 68.58 | 96.07 |
| Ours | H36M-Aug, MI, COCO, LSP, LSPET, MPII | Twin-SVT-B | 47.70 | **79.16** | 66.53 | **95.03** |
| Ours | EFT-[COCO, LSPET, MPII], H36M, SPIN-MI | Twin-SVT-B | **47.31** | 81.90 | **64.19** | 96.56 |
| Ours | H36M, MI, EFT-COCO | HrNet-W32 | 48.08 | 83.16 | 66.01 | 100.59 |
| Ours | H36M, MI, EFT-COCO | Twin-SVT-B | 48.27 | 84.39 | 64.72 | 99.61 |
| Ours | H36M, MuCo, EFT-COCO | Twin-SVT-B | 47.76 | 80.03 | 64.43 | 98.07 |
| Ours | EFT-[COCO, LSPET, PT, OCH] H36M, MI | Twin-SVT-B | 49.33 | 83.13 | 66.29 | 99.73 |

are becoming saturated in the PA-MPJPE range of 50+ $mm$, making it difficult to evaluate how close the field is to fully robust and general solutions. Through this study, we manage to attain a PA-MPJPE of 47.68 $mm$ using the same backbone and dataset selection as PARE [33], which reports 50.9 $mm$ with a more sophisticated algorithm. Keeping model capacity and dataset selection similar to HMR (76.7 $mm$) [30] and SPIN (59.2 $mm$) [35], we reach 51.66 $mm$. Additionally, we achieve 48.77 $mm$ using HMR's original dataset and partition which does not contain any EFT or SPIN fittings. With more robust dataset choices following [33], our best model obtains 47.31 $mm$ without fine-tuning on 3DPW train set. We hope our competitive results could propel the community to focus on newer algorithms and draw attention away from different training settings in the future.

## 2 Preliminaries

**Base model.** The origin of many mesh recovery works [35, 12, 37, 32, 33, 62, 59, 48] can be traced back to HMR [30]. It adopts a neural network to regress the parameters of a SMPL body [30], which is a differentiable function that maps pose parameters $\theta$ and shape parameters $\beta$ to a triangulated mesh with 6980 vertices. Following this study, subsequent works have been built upon HMR to further enhance the recovery performance. For instance, some solutions are proposed to improve the robustness by adding an optimization loop [35], estimating camera parameters [34] or using probabilistic estimation to derive the pose [37]; some works also extend HMR to predict the appearance (e.g., HMAR [63]) or temporal dimension (e.g., HMMR [60], VIBE [32], MEVA [48]). We benchmark HMR as it has also been widely used as the baseline in many studies [58, 29, 5, 12]. In Section 6, we also demonstrate benchmarking results on other algorithms.

**Evaluation.** We follow the widely adopted evaluation protocol in [30, 35]. Performance is measured in terms of recovery errors (PA-MPJPE) in $mm$. A smaller PA-MPJPE value indicates better recovery

Table 2: **HMR model performance when trained on individual datasets. For PROX and MuPoTs-3D, only 2D keypoints are used for training. P: person-person occlusion O: person-object occlusion.**

| Training dataset | Annotation type | Env. | # Samples | # Subjects | # Scenes | # Cam | Occ. | PA-MPJPE↓ | MPJPE↓ | PA-PVE↓ | PVE↓ |
|---|---|---|---|---|---|---|---|---|---|---|---|
| PROX [20] * | 2DKP | Indoor | 88484 | 11 | 12 | - | O | 84.69 | 147.93 | 109.85 | 177.01 |
| COCO-Wholebody [25] | 2DKP | Outdoor | 40055 | 40055 | - | - | - | 85.27 | 157.13 | 107.44 | 176.49 |
| Instavariety [31] | 2DKP | Outdoor | 2187158 | >28272 | - | - | - | 88.93 | 151.22 | 122.51 | 184.15 |
| COCO [45] | 2DKP | Outdoor | 28344 | 28344 | - | - | - | 93.18 | 197.47 | 122.05 | 238.30 |
| MuPoTs-3D [52] * | 2DKP | Outdoor | 20760 | 8 | - | 12 | - | 95.83 | 190.88 | 121.58 | 241.89 |
| LIP [17] | 2DKP | Outdoor | 25553 | 25553 | - | - | - | 96.47 | 198.65 | 123.78 | 241.98 |
| MPII [1] | 2DKP | Outdoor | 14810 | 14810 | 3913 | - | - | 98.18 | 228.90 | 128.95 | 246.61 |
| Crowdpose [39] | 2DKP | Outdoor | 13927 | - | - | - | P | 99.97 | 207.03 | 136.45 | 240.35 |
| Vlog People [31] | 2DKP | Outdoor | 353306 | 798 | 798 | - | - | 100.38 | 201.69 | 135.86 | 245.75 |
| PoseTrack (PT) [2] | 2DKP | Outdoor | 5084 | 550 | 550 | - | - | 105.30 | 229.44 | 141.58 | 270.99 |
| LSP [26] | 2DKP | Outdoor | 999 | 999 | - | - | - | 111.45 | 247.29 | 154.63 | 293.38 |
| AI Challenger [77] | 2DKP | Outdoor | 378374 | - | - | - | - | 111.66 | 255.35 | 147.40 | 305.342 |
| LSPET [27] | 2DKP | Outdoor | 9427 | 9427 | - | - | - | 112.26 | 328.98 | 139.79 | 387.05 |
| Penn-Action [88] | 2DKP | Outdoor | 17443 | 2326 | 2326 | - | - | 114.53 | 370.03 | 144.84 | 447.89 |
| OCHuman (OCH) [86] | 2DKP | Outdoor | 10375 | 8110 | - | - | P,O | 130.55 | 262.62 | 157.68 | 315.87 |
| MuCo-3DHP (MuCo) [52] | 2DKP/ 3DKP | Indoor | 482725 | 8 | - | 14 | P | 78.05 | 144.25 | 101.19 | 164.02 |
| MPI-INF-3DHP (MI) [51] | 2DKP/ 3DKP | Indoor | 105274 | 8 | 1 | 14 | - | 107.15 | 232.47 | 140.74 | 274.58 |
| 3DOH50K (OH) [87] | 2DKP/ 3DKP | Indoor | 50310 | - | 1 | 6 | O | 114.48 | 302.57 | 248.07 | 346.12 |
| 3D People [61] | 2DKP/ 3DKP | Indoor | 1984640 | 80 | - | 4 | - | 108.27 | 229.89 | 127.21 | 253.38 |
| AGORA [58] | 2DKP/ 3DKP/ SMPL | Indoor | 100015 | >350 | - | - | P,O | 77.94 | 140.64 | 98.40 | 161.91 |
| SURREAL [75] | 2DKP/ 3DKP/ SMPL | Indoor | 1605030 | 145 | 2607 | - | - | 110.00 | 291.17 | 142.53 | 372.78 |
| Human3.6M (H36M) [23] | 2DKP/ 3DKP/ SMPL | Indoor | 312188 | 9 | 1 | 4 | - | 124.55 | 286.12 | 170.57 | 326.40 |
| EFT-COCO [29] | 2DKP/ SMPL | Outdoor | 74834 | 74834 | - | - | - | 60.82 | 96.20 | 78.28 | 114.61 |
| EFT-COCO-part [29] | 2DKP/ SMPL | Outdoor | 28062 | 28062 | - | - | - | 67.81 | 110.00 | 86.77 | 128.62 |
| EFT-PoseTrack [29] | 2DKP/ SMPL | Outdoor | 28457 | 550 | - | - | - | 75.17 | 127.87 | 96.61 | 149.14 |
| EFT-MPII [29] | 2DKP/ SMPL | Outdoor | 14667 | 3913 | - | - | - | 77.67 | 132.46 | 97.97 | 150.55 |
| UP-3D [38] | 2DKP/ SMPL | Outdoor | 7126 | 7126 | - | - | - | 86.92 | 161.61 | 109.51 | 181.00 |
| MTP [55] | 2DKP/ SMPL | Outdoor | 3187 | 3187 | - | - | - | 87.03 | 191.08 | 110.43 | 227.36 |
| EFT-OCHUMAN [29] | 2DKP/ SMPL | Outdoor | 2495 | 2495 | - | - | P,O | 93.44 | 187.38 | 123.03 | 216.06 |
| EFT-LSPET [29] | 2DKP/ SMPL | Outdoor | 2946 | 2946 | - | - | - | 100.53 | 208.90 | 128.77 | 240.69 |
| 3DPW [76] | SMPL | Outdoor | 22735 | 7 | - | - | - | 89.36 | 168.98 | 115.09 | 207.98 |

performance. Our goal is to infer accurate pose $\theta$ and shape parameters $\beta$, which are later taken as input for parametric human models to get joint locations. This metric has already implied the evaluation of human shape and mesh [43, 73, 35, 44]. [83, 41] pointed out that PA-MPJPE is not perfect, thus we have add more metrics such as PVE, PA-PVE and MPJPE.

We adopt the 3DPW [75] test set for evaluation without any fine-tuning on its training set (*Protocol 2*)[2]. In Section 6, we also provide evaluations on other test sets and show that 3DPW is a representative benchmark. This outdoor dataset is often used as the main or only benchmark [30, 35, 71, 12, 29, 32, 33] to assess real-world systems under a wide variety of in-the-wild conditions. We also evaluate the indoor H36M test set [23]. The results can be found in the appendix, which gives the same conclusions as 3DPW. We train the model for 100 epochs[3] and evaluate its performance in each epoch. After this, the best PA-MPJPE is reported. We perform our benchmarking from three perspectives - datasets (Section 3), backbones (Section 4) and training strategies (Section 5).

## 3 Benchmarking Training Datasets

Training datasets play an important role in determining mesh recovery accuracy. Table 12 in the appendix summarises the datasets used in various algorithms. Many works train on their own unique combinations of datasets determined heuristically [33, 34, 29, 40, 12]. This makes it hard to attribute performance gains to the proposed algorithm or to the handpicked selection of datasets, and necessitates benchmarks on different dataset choices. We provide a systematic and comprehensive evaluation of the impact of training datasets on the HMR performance. Our benchmarks involve not only the datasets used in prior mesh recovery works, but also the newest ones (e.g., PROX [20], AGORA [58]) as well as those commonly used in 2D/3D pose estimation (e.g., LIP [17], Crowdpose [39], AI Challenger [77], Penn-Action [88], MuCo-3DHP [52], etc.). We consider different factors in the training datasets that can affect the model performance, which are rarely investigated previously.

### 3.1 Dataset Attributes

Different datasets may exhibit different attributes, which are critical for the model performance. To easily analyze their impacts, we use each dataset from our collection to train the HMR model, and test its performance. Table 2 summarizes the attributes and the corresponding performance.

**Non-critical attributes**. Joo et al. [29] suggested that there exists an indoor-outdoor domain gap, where models trained on outdoor datasets do not perform well on indoor test datasets, and vice versa. However, our comprehensive benchmarks reveal that not all datasets' performance can be explained by the indoor-outdoor domain gap, calling for a more careful analysis of underlying factors.

---

[2]Some works [34, 80, 48, 43, 68, 19] also adopt the 3DPW training set during training (*Protocol 1*). In general, using the 3DPW training data improves the performance but it is not a universal practice.

[3]All models were trained with 8 Tesla V100 GPUs.

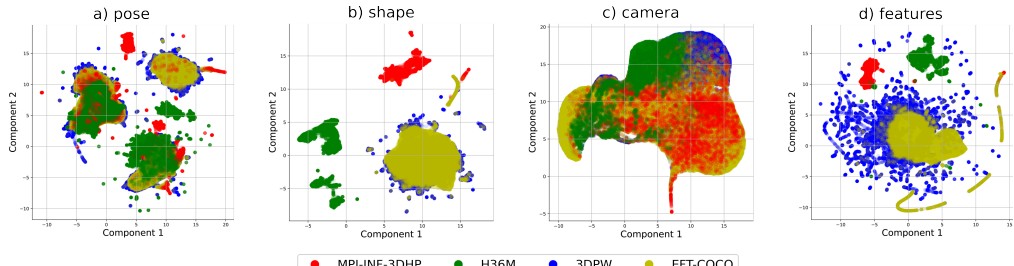

Figure 1: **Distributions of the four attributes in four datasets (better viewed in color)**.

For instance, several notable indoor training datasets (*e.g.*, PROX, MuCo-3DHP) outperform many outdoor datasets, and result in high-performing models on the outdoor 3DPW test dataset. Similarly, we observe that some indoor training datasets (*e.g.*, MPI-INF-3DHP, 3DOH50K) give a really poor performance on the indoor H36M test dataset, as shown in the appendix. In addition, we find weak correlations between the number of data points and model performance. For instance, COCO with 10x fewer data points outperforms H36M [23] on the 3DPW test set.

**Critical attributes.** There are some attributes that can heavily impact the model performance, such as human poses, body shape (height, limb length), scenes, lighting, occlusion (self, people, environment), annotation types (2D/3D keypoints, SMPL) and camera characteristics (angles) [58, 5, 73, 64, 29, 4]. High similarities of these attributes between the training and test datasets can yield better performance.

To validate these, we adopt a well-trained HMR model to estimate the distributions of four attributes: 1) pose $\theta \in R^{69}$, 2) shape $\beta \in R^{10}$ and 3) camera translation $t^c \in R^3$ obtained from the head, and 4) features $f \in R^{2048}$ obtained from the ResNet-50 backbone. Fig. 1 visualizes the results with the UMAP dimension reduction technique [50] for four selected datasets: COCO, 3DPW, H36M and MPI-INF-3DHP. We have the following observations. First, COCO has a large variety of these attributes, which considerably overlap with those of 3DPW. This explains why training with COCO gives satisfactory performance on 3DPW. Second, H36M lacks diversity in poses (Fig. 1a) and has distinctly different distributions of features (Fig. 1d) and shape (Fig. 1b) from 3DPW, possibly due to the limited number of subjects (9) and scenes (1) (Table 2). In addition, H36M's shape and camera distribution differ from MPI-INF-3DHP. Therefore, training with either 3DPW or MPI-INF-3DHP has poor performance on H36M. H36M benchmarks and extra visualization of the attribute distributions for other datasets (Figs. 12 - 15) can be found in the appendix.

Notably, the indoor datasets that perform well on outdoor 3DPW benchmarks are designed with considerable person-person (MuCo-3DHP) and person-object occlusion (PROX) (Fig. 6 in the appendix). This suggests that occlusion can be a more important factor that predominates the background (see Appendix C for more details).

To demonstrate the importance of the SMPL fitting mechanism, we compare EFT datasets with and without SMPL annotation, as shown in Table 3. We observe that EFT fittings can reduce the PA-MPJPE by over 20 $mm$ for different datasets. This is consistent with the findings from [29, 5] that SMPL parameters ($\theta$ and $\beta$) provide stronger supervision signals compared to 2D and 3D keypoints. Cai et al. [5] put forward the reason that strong supervision initiates the gradient flow that reaches the learnable SMPL parameters in the shortest possible route.

Table 3: **HMR model performance with EFT datasets**.

| Dataset | w/ SMPL | w/o SMPL |
|---|---|---|
| EFT-COCO | 60.82 | 94.42 |
| EFT-COCO-Part | 67.81 | 101.65 |
| EFT-PoseTrack | 75.17 | 103.10 |
| EFT-MPII | 77.66 | 99.87 |
| EFT-OCHuman | 94.01 | 121.68 |
| EFT-LSPET | 100.53 | 134.62 |

> **Remark #1:** The indoor/outdoor settings or number of data points are not strong indicators for the model performance. Some attributes (e.g., human pose and shape, camera characteristics, backbone features) are more critical, and having high diversities (leading to considerable overlap between the training and test sets distributions) can give more satisfactory results. Occlusion (person-person or person-object) and SMPL fittings can also help boost recovery accuracy.

## 3.2 Combination of Multiple Datasets

It is a common practice to train the mesh recovery model with multiple datasets of different domains and annotation types. Past works select the datasets empirically. We argue that different combinations of datasets can lead to a vast fluctuation in performance. We explore their impacts from two directions.

Table 5: **HMR model performance when trained with different contribution configurations of six datasets. (Left) Direct partition. (Right) Reweight samples.**

| Partition | | | | | | PA-MPJPE↓ |
| --- | --- | --- | --- | --- | --- | --- |
| H36M | MI | LSPET | LSP | MPII | COCO | |
| 0.35 | 0.15 | 0.10 | 0.10 | 0.10 | 0.20 | 64.55 |
| 0.10 | 0.10 | 0.10 | 0.05 | 0.15 | 0.50 | 61.66 |
| 0.20 | 0.10 | 0.10 | 0.05 | 0.15 | 0.40 | 61.23 |
| 0.40 | 0.20 | 0.10 | 0.10 | 0.10 | 0.10 | 66.33 |
| 0.17 | 0.17 | 0.17 | 0.17 | 0.17 | 0.17 | 63.10 |

| Weighting | | | | | | PA-MPJPE↓ |
| --- | --- | --- | --- | --- | --- | --- |
| H36M | MI | LSPET | LSP | MPII | COCO | |
| 0.17 | 0.17 | 0.17 | 0.17 | 0.17 | 0.17 | 63.25 |
| 0.10 | 0.10 | 0.10 | 0.05 | 0.15 | 0.50 | 62.43 |
| 0.20 | 0.10 | 0.10 | 0.05 | 0.15 | 0.40 | 62.47 |
| 0.20 | 0.10 | 0.15 | 0.10 | 0.15 | 0.40 | 63.51 |
| 0.35 | 0.15 | 0.10 | 0.10 | 0.10 | 0.20 | 64.93 |

**Selection of datasets.** We first evaluate different combinations of training datasets, as shown in Table 4. Particularly, *Mix 2* follows the selection in DSR [12] and EFT [29] while *Mix 6* follows that in PARE [33]. We have several observations. First, the selection of the training sets has high impacts on the model per-

Table 4: **HMR model performance when trained with different combinations of datasets.**

| Mix | Datasets | PA-MPJPE↓ | MPJPE↓ | PA-PVE↓ | PVE↓ |
| --- | --- | --- | --- | --- | --- |
| 1 | H36M, MI, COCO | 66.14 | 115.19 | 89.04 | 135.68 |
| 2 | H36M, MI, EFT-COCO | 55.98 | 91.68 | 73.17 | 107.39 |
| 3 | H36M, MI, EFT-COCO, MPII | 56.39 | 94.56 | 74.88 | 111.40 |
| 4 | H36M, MuCo, EFT-COCO | 53.90 | 87.76 | 71.10 | 104.59 |
| 5 | H36M, MI, COCO, LSP, LSPET, MPII | 64.55 | 109.73 | 86.62 | 128.93 |
| 6 | EFT-[COCO, MPII, LSPET], SPIN-MI, H36M | 55.47 | 90.77 | 72.78 | 107.08 |
| 7 | EFT-[COCO, MPII, LSPET], MuCo, H36M, PROX | 52.96 | **86.00** | **70.34** | 104.49 |
| 8 | EFT-[COCO, PT, LSPET], MI, H36M | 55.97 | 91.34 | 73.63 | 107.90 |
| 9 | EFT-[COCO, PT, LSPET, OCH], MI, H36M | 55.59 | 89.91 | 73.20 | 106.17 |
| 10 | PROX, MuCo, EFT-[COCO, PT, LSPET, OCH], UP-3D, MTP, Crowdpose | 57.80 | 96.41 | 75.01 | 113.55 |
| 11 | EFT-[COCO, MPII, LSPET], MuCo, H36M | **52.54** | 86.68 | 70.63 | **103.07** |

formance, even more critical than the training algorithms. For instance, for *Mix 2*, we obtain a PA-MPJPE of 55.98 $mm$ using the HMR base model, while DSR and EFT report 54.1 $mm$ and 54.7 $mm$ respectively, with more sophisticated algorithms. The performance difference is minor compared to that with different combinations of datasets. Similarly, for *Mix 6*, our HMR base model gets 55.47 $mm$ whereas PARE reports 52.3 $mm$. We observe that some prior works compare their model performance with algorithms trained on vastly different dataset mixes, which is arguably unfair. The lack of a defined and consistent combination of training datasets hinders the direct comparison of different algorithms' performance. Through our benchmarking, we provide the community with new baselines on some of the commonly used dataset combinations.

Second, it is not necessary to include more datasets. From Table 4, we observe that *Mix 10* does not perform as well as other mixes with fewer datasets. Involving more datasets could harm the model accuracy. It is recommended to select the optimal combination of datasets rather than prioritizing the quantity. For instance, we discover that the involvement of EFT (especially EFT-COCO) datasets can boost the performance, which should be strongly considered as the baselines (Table 4).

It is worth noting that we heuristically select the datasets for benchmarking. We emphasize that this selection process is not directionless. From our analysis, a good overlap in train-test distributions of features (e.g., camera, pose, shape, backbone features) would help to achieve good performance. We could then select the top $N$ datasets that would cover a wide distribution. In addition, we identify attributes that makes a dataset effective for training, such as the presence of SMPL annotations, and few datasets currently afford it. These findings help us make informed choices on dataset selection. How to automatically select the datasets will be our future work.

**Dataset contributions.** In addition to the selection of datasets, the relative contribution of each dataset in the combination is also important. Unfortunately, no prior works consider this factor. Basically, there are two approaches to adjust the dataset contribution. The first one is to set the partitions (i.e., probability that a dataset is "seen" during training) of these datasets with pre-defined ratios [30]. The second is to maintain the same partition and reweight samples from different datasets, similar to prior methods of weighting valuable samples [65, 69]. Table 5 shows the model performance with different contribution configurations using 6 datasets in [30]. Our observations include: (1) Setting different contributions for different datasets can indeed alter the model performance. A careful configuration can bring a rather large improvement. It is important to increase the contributions of critical datasets that can benefit the training (e.g., COCO in our case). (2) Under the same contribution configurations, the approach of directly altering the partitions is more effective than reweighting the samples.

> **Remark #2:** The selection of datasets and their relative contributions are important factors to determine the model performance. To fairly evaluate and compare the impact of other factors (e.g., training algorithms), it is crucial to keep the same dataset combination configuration, which is usually ignored by prior works. To get a good baseline model, it is suggested to adopt more critical datasets and increase their contribution during training.

## 3.3 Annotation Quality

Many algorithms use pseudo-annotations during training (Table 12 in the appendix). We assess how the annotation quality affects training. To this end, we generate datasets with controlled noise to reflect different magnitudes of corruption in real scenarios. We investigate the following aspects.

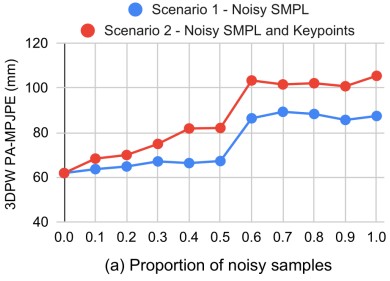 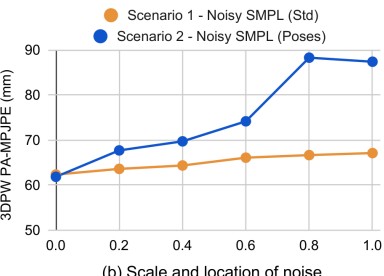

Figure 2: **HMR model performance with different types of noisy training data.**

**Proportion of noisy samples.** We inject noise to different ratios of samples. We consider two scenarios: (1) only SMPL annotations is noisy, which might occur when challenging poses are wrongly fitted; (2) both keypoints and SMPL annotations are noisy as incorrect keypoint estimation leads to erroneous fittings [29] (Fig. 7 in the appendix).

Fig. 2a shows the model performance on the 3DPW test set with different ratios of noisy samples in the above two scenarios. For scenario (1), the errors remain low under small portions of noisy SMPL annotations (<50%). The trained model can generalize and learn with such noisy samples. When the amount of noisy SMPL annotations overwhelms the clean ones, the errors increase sharply. For scenario (2), when we add noisy keypoints on top of noisy SMPL, we obtain large increments in the recovery errors. It also increases significantly with more than 50% noisy samples. A plausible reason is that when there are only noisy SMPL annotations, the clean keypoints can still provide supervision to keep errors low. However, when both keypoints and SMPL are noisy, the errors are dire.

**Scale and location of noise.** We further consider two more scenarios about the controlled noise (Fig. 8 in the appendix). (1) We inject noise to the SMPL of all poses and vary its scales (i.e. Simple Gaussian Noise with different standard deviations following [22, 14]). The generated poses are still realistic. (2) We observe that fitted poses of certain body parts (i.e. feet and hand) tend to be less accurate in existing fittings. We simulate cases for wrongly fitted body parts by replacing a percentage of pose parameters (body parts) with random noise.

Fig. 2b shows the model performance, where we add noise of different standard deviations to all pose parameters (scenario 1), or totally random noise to different ratios of the pose parameters (scenario 2). When we increase the noise scale, the errors increase slightly and remain low (<70). However, when a portion of body part is replaced with random noise, errors increase by a large margin. This shows that clean SMPL annotations are important, but slightly noisy SMPL within realistic realms can still be useful for training. SMPL fittings should be reasonably accurate, but need not be perfect.

> **Remark #3:** Noisy data samples can harm the model performance, especially when the ratio of samples is higher, or both the SMPL annotation and keypoints are compromised. Slightly noisy SMPL still helps training.

## 4 Benchmarking Backbone Models

**Model architecture.**

Following Kanazawa et al. [30], ResNet-50 [21] is the default backbone in many mesh recovery works [29, 33, 32, 35]. More recently, Kocabas et al. [33] adopted HRNet-W32 [70] and attributed the performance gains to its ability to produce more robust high-resolution representations. We further consider other architectures.

Table 6: **HMR model performance with different backbone architectures**.

| Backbone | Params (M) | FLOPs (G) | PA-MPJPE↓ | MPJPE↓ | PA-PVE↓ | PVE↓ |
|---|---|---|---|---|---|---|
| ResNet-50 [21] | 28.79 | 4.13 | 64.55 | 112.34 | 89.05 | 130.41 |
| ResNet-101 [21] | 47.78 | 7.83 | 63.36 | 112.67 | 82.65 | 129.71 |
| ResNet-152 [21] | 63.42 | 11.54 | 62.13 | 107.13 | 81.45 | 123.95 |
| HRNet-W32 [70] | 36.69 | 11.05 | 64.27 | 108.32 | 82.86 | 122.36 |
| EfficientNet-B5 [72] | 33.62 | 0.03 | 65.16 | 118.15 | 83.88 | 144.23 |
| ResNext-101 [78] | 91.39 | 16.45 | 64.95 | 114.43 | 87.26 | 130.28 |
| Swin [46] | 51.72 | 32.48 | 62.78 | 110.42 | 84.88 | 137.26 |
| ViT [11] | 91.07 | 11.29 | 62.81 | 111.46 | 84.01 | 127.22 |
| Twin-SVT [9] | 59.27 | 8.35 | 60.11 | **100.75** | **79.00** | **121.05** |
| Twin-PCVCT [9] | 47.02 | 6.45 | **59.13** | 103.85 | 80.62 | 123.93 |

Particularly, we compare different variations of CNN-based models (ResNet-101, ResNet-152, HR-

Net, EfficientNet [72], ResNext [78]), as well as the latest transformer-based architectures (ViT [11], Swin [46], Twins (-SVT and -PCVCT) [9]).

Table 6 reports the performance of the HMR model trained with different backbone architectures. First, increasing the backbone capacity allows deeper feature representations to be learned, yielding performance gains. For instance, the PA-MPJPE is reduced when we switch the backbone model from ResNet-50 to ResNet-152. This is consistent with the findings in [5]. Second, transformer-based backbones are superior to CNN-based backbones, achieving lower PA-MPJPEs and similar FLOPs under comparable parameters (Table 6). They are capable of mining rich structured patterns, which are especially essential for learning from different data sources. This contradicts the discoveries in [5], which did not find the advantage of vision transformers over CNN-based ones.

**Weight initialization.** It is common and computationally efficient to build the HMR model based on a pre-trained backbone. Initialization of the backbone model weights has a significant impact on the HMR model performance. PARE [33] is the first work to use weights from a pose estimation task. It initializes the weights of the HRNet-W32 backbone from a pose estimation model trained on MPII. The initialized model is further finetuned on EFT-COCO for 175K steps before training on *Mix 6*. Kocabas et al. [33] noted that this strategy accelerates the model convergence and reduces overall training time. However, this study does not provide ablation studies to explore the effect of using pretrained weights from a pose estimation model.

To disclose the impact of weight initialization, we systematically benchmark strategies where the backbone weights are pre-trained with ImageNet, or from pose estimation models trained over MPII or COCO. The results are reported in Table 7. First, we observe that transferring knowledge from a strong pose estimation model is sufficient to achieve large improvement gains

Table 7: **HMR model performance with different weight initializations.**

| Backbone | Mixed Datasets | Dataset for weight initialization | | |
|---|---|---|---|---|
| | | ImageNet | MPII | COCO |
| ResNet-50 | HMR/SPIN | 64.55 | 60.60 | 57.26 |
| HRNet-W32 | HMR/SPIN | 64.27 | 55.93 | 54.47 |
| Twin-SVT-B | HMR/SPIN | 60.11 | 56.80 | 52.61 |
| HRNet-W32 | PARE | 54.84 | 51.50 | 49.54 |

without having to fine-tune on EFT-COCO, as done in PARE. In Table 7, with the HRNet-W32 backbone and weights initialized from MPII, we can already achieve a PA-MPJPE of $51.5\ mm$, which is very close to the error of $50.9\ mm$ reported by PARE [33]. The effectiveness of such a pretrained backbone suggests that features learnt from pose estimation tasks are robust and complementary for mesh recovery tasks. Second, the choice of the pose estimation dataset for weight initialization is also vital. Regardless of the backbone variants, pretraining the backbone with COCO gives better performance than MPII for different training dataset mixes and backbone architectures.

> **Remark #4:** The backbone architecture and weight initialization are vital for the HMR performance. Optimal configurations comprise of transformer-based backbones with weights initialized from a strong pose estimation model trained on in-the-wild datasets.

## 5 Benchmarking Training Strategies

### 5.1 Augmentation

Various augmentation methods have been adopted for mesh recovery. SPIN [33] utilized rotation, flip and color noising. PARE [33] and BMP [81] added synthetic occlusion by compositing a random nonhuman object to the image. BMP [85] made it keypoint-aware by occluding randomly selected keypoints. Georgakis et al. [15] controlled the extent of occlusion by varying the pattern (oriented bars, circles, rectangles) size. Mehta et al. [52] created inter-person overlap in the datasets. Other than occlusion, crop augmentation has also been applied to better reconstruct highly truncated people [66, 29, 33]. Augmentation has also been used to bridge the synthetic-to-real domain gap [59, 10, 79, 67]. However, the above studies adopt different configurations and benchmarks, and it is hard to get general conclusions about the effect of different augmentations.

We systematically evaluate and compare 9 image-based augmentations over different training and test datasets. Specifically, we re-implement common augmentation techniques in prior mesh recovery works, such as random occlusion (or hard erasing) [40, 89], synthetic occlusion [33] and crop augmentation [33, 29]. Besides, we also adopt popular augmentations from person re-identification and pose estimation tasks, such as soft erasing [7], self-mixing [7], photometric distortion [3], coarse and grid dropouts [3]. Fig. 3 visualizes the augmented results by different techniques.

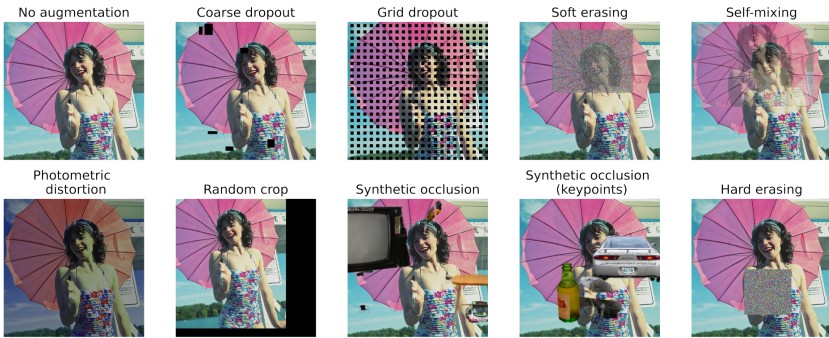

| No augmentation | Coarse dropout | Grid dropout | Soft erasing | Self-mixing |
| --- | --- | --- | --- | --- |
| Photometric distortion | Random crop | Synthetic occlusion | Synthetic occlusion (keypoints) | Hard erasing |

Figure 3: **Visualisation of augmented samples.**

Table 8: **HMR model performance on test sets of 3DPW [76], EFT-LSPET [29], EFT-OCH [29] and H36M [23] and validation set of EFT-COCO [29] when trained on H36M and EFT-COCO with different augmentations. Blue: Augmentation improves the performance. Red: Augmentation harms the performance. Bold: best in column. Underline: second best in column.**

| Augmentation | H36M-train | | | | | EFT-COCO-train | | | | |
| --- | --- | --- | --- | --- | --- | --- | --- | --- | --- | --- |
| | 3DPW↓ | LSPET↓ | OCH↓ | COCO↓ | H36M↓ | 3DPW↓ | LSPET↓ | OCH↓ | COCO↓ | H36M↓ |
| No augmentation | 124.55 | 207.45 | 161.77 | 165.03 | 53.73 | 62.37 | 131.71 | **115.50** | 114.59 | 118.39 |
| Hard erasing | 107.03 | 201.16 | 153.87 | 147.00 | 51.70 | 64.77 | 136.90 | 118.93 | 115.61 | 120.78 |
| Soft erasing | 107.10 | 193.33 | 149.93 | 143.51 | 47.77 | 65.70 | 139.21 | 118.29 | **100.01** | 131.09 |
| Self mixing | **101.10** | 191.70 | **136.68** | **132.17** | **45.12** | 63.98 | 133.18 | 118.30 | 125.32 | 104.37 |
| Photometric distortion | 113.53 | 190.60 | 155.57 | 153.95 | 48.45 | 62.07 | **128.45** | 116.47 | 112.88 | 118.92 |
| Random crop | 110.08 | 205.91 | 150.33 | 147.27 | 52.53 | 71.21 | 148.80 | 124.14 | 104.43 | 100.43 |
| Synthetic occ. | 101.96 | 221.79 | 146.44 | 143.32 | 48.27 | 63.94 | 135.00 | 116.25 | 103.36 | 107.14 |
| Synthetic occ. (kp) | 107.68 | 215.34 | 153.90 | 145.70 | 52.26 | 71.35 | 142.93 | 121.34 | 100.90 | 103.79 |
| Grid dropout | 117.45 | 208.49 | 161.69 | 158.27 | 57.20 | 66.65 | 139.71 | 118.89 | 100.52 | 103.07 |
| Coarse dropout | 124.99 | 202.74 | 162.50 | 159.48 | 50.61 | 62.78 | 128.61 | 116.58 | 119.70 | 127.92 |

We consider two individual training datasets with distinct characteristics: the indoor H36M set and outdoor EFT-COCO. We apply different augmentations to both datasets to train the HMR models, before evaluating them on five test datasets: 3DPW, EFT-LSPET, EFT-OCHuman, EFT-COCO and H36M. Table 8 reveals the distinct effects of augmentation on these two training datasets. (1) For H36M, almost all the augmentations help the trained model achieve lower errors across outdoor test sets, and self-mixing is the most effective solution. This implies that augmentation can help bridge the indoor-outdoor domain gap and prevent overfitting. In the appendix, Fig. 9 compares the training curves with and without augmentation to confirm this conclusion. Fig. 10 shows the distribution of camera features after applying self-mixing, which overlaps substantially with the predicted features obtained from a robust model that performs well on 3DPW-test. (2) For EFT-COCO, we observe that robust augmentations seldom improve the model performance on the test sets of 3DPW, EFT-LSPET and EFT-OCHuman, with the exception of EFT-COCO-val. This is consistent with the findings in Joo et al. [29]: EFT-COCO-train already includes many robust samples with severe occlusions. Adding more extensive augmentation can harm the model performance.

> **Remark #5:** The effect of data augmentations highly depends on the characteristics of the training dataset. Their benefits are more obvious when the training sets contain less diverse and robust samples. When combining multiple datasets during training, we can selectively apply data augmentations to different datasets based on their characteristics.

## 5.2 Training Loss

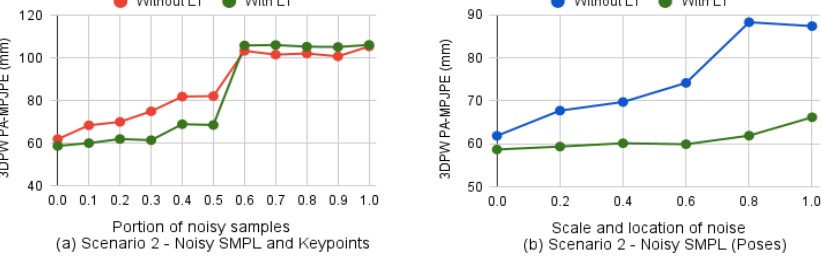

Figure 4: **HMR model performance with and without L1 loss under different (a) proportions of noisy SMPL and keypoints; (b) ratios of noisy pose parameters.**

Prior works commonly adopt the MSE loss in pose estimation involving keypoints [13, 56]. In HMR, regression of keypoints and SMPL parameters are supervised with the MSE loss. We experiment with alternative settings where a different loss is applied. As the L1 loss function measures the magnitude of the error but does not consider the direction, it is insensitive to outliers. Under certain assumptions, Ghosh et al. [16] theoretically demonstrated that L1 can be robust against noisy labels. Inspired by this, we use L1 in place of MSE loss for the regression of keypoints and SMPL parameters in HMR. We note that this replacement can improve the model from two directions. First, it helps to tackle noisy samples. Fig. 4 compares the HMR model performance with MSE and L1 loss functions under different scales of SMPL noise, and proportions of noisy keypoints and SMPL annotations, respectively. We find that L1 loss can make the model more robust to noise.

Second, L1 loss improves model performance under the multi-dataset setting. Table 9 compares the performance with and without L1 loss trained on different dataset combinations. We observe that L1 loss can bring significant performance gains. In particular, applying L1 loss to the dataset configurations in SPIN [35] reduces the errors from 64.55 $mm$ to 58.20 $mm$. We also note that

Table 9: **HMR model performance with and without L1 loss under multi-dataset setting**.

| Mix | Datasets | w/oL1 | w/L1 |
|---|---|---|---|
| 1 | H36M, MI, COCO | 66.14 | **57.01** |
| 2 | H36M, MI, EFT-COCO | 55.98 | **55.25** |
| 5 | H36M, MI, COCO, LSP, LSPET, MPII | 64.55 | **58.20** |
| 6 | EFT-[COCO, MPII, LSPET], SPIN-MI, H36M | 55.47 | **53.62** |
| 8 | EFT-[COCO, PT, LSPET], MI, H36M | 55.97 | **53.43** |
| 7 | EFT-[COCO, MPII, LSPET], MuCo, H36M, PROX | 53.44 | **52.93** |
| 11 | EFT-[COCO, MPII, LSPET], MuCo, H36M | **52.54** | 53.17 |

the gains from L1 loss becomes smaller when the dataset selection is more optimal (*Mix 2, 6, 8*).

> **Remark #6:** Prior works adopt MSE loss for regression of keypoints and SMPL parameters. Using L1 loss instead can not only improve the model's robustness against noisy samples, but also enhance the model performance, especially when the selected datasets are not optimal.

## 6 Benchmarking Other Algorithms and Test Sets

In the previous benchmarking experiments, we choose the HMR algorithm and 3DPW test set. Our evaluation methodology and conclusions are general to other algorithms and test sets as well. In this section, we demonstrate some experiments to validate their generalization.

**Other algorithms.** In addition to HMR, we consider some other popular algorithms (SPIN [35], GraphCMR [36], PARE [33], Graphormer [44]). Table 10 reports the model performance for different algorithms and configurations[4]. Table 16 in Appendix considers different dataset mixes and backbones. We can easily observe that similar to HMR, high-quality models for other algorithms are also established with L1 loss, weight initialisation from COCO pose estimation model, and selective augmentation.

Table 10: **Model performance (3DPW-test PA-MPJPE in $mm$) when trained with different recommended strategies of L1 loss, weight initialisation from COCO pose estimation model, and selective augmentation.**

| Algorithms | Datasets | Backbones | Initialisation | Normal | L1 | L1+COCO | L1+COCO+Aug |
|---|---|---|---|---|---|---|---|
| HMR | H36M, MI, COCO, MPII, LSP, LSPET | ResNet-50 | ImageNet | 64.55 | 58.20 | 51.80 | 51.66 |
| SPIN | H36M, MI, COCO, MPII, LSP, LSPET | ResNet-50 | HMR (ImageNet) | 59.00 | 57.08 | 51.54 | 50.69 |
| GraphCMR | COCO, H36M, MPII, LSPET, LSP, UP3D | ResNet-50 | ImageNet | 70.51 | 67.20 | 61.74 | 60.26 |
| PARE | EFT-[COCO, LSPET, MPII], H36, MI | HRNet-W32 | ImageNet | 61.99 | 61.13 | 59.98 | 58.32 |
| Graphormer | H36M, COCO, UP3D, MPII, MuCo | HRNet-W48 | ImageNet | 63.18 | 63.47 | 59.66 | 58.82 |

**Other test sets.** In addition to the 3DPW, other works have evaluated on MuPoTs-3D-test [81], AGORA-test [42], MPI-INF-3DHP-test [40] and Joo et al. [29] suggested EFT-OCHuman-test and EFT-LSPET-test for more challenging benchmarks. For comprehensive benchmarking, we include 7 more test sets: (H36M, AGORA validation, MPI-INF-3DHP test, EFT-COCO validation, MuPots-3D test, EFT-OCHuman test, EFT-LSPET test) for evaluations. We run dataset benchmarks on all selected test sets and compute the correlation between the model performance on different test sets. The results are shown in Table 11. We find good correlations between the performance on 3DPW with that on other test sets. This indicates that 3DPW is a fairly good benchmark, and evaluations on

---

[4]Our baseline models for HMR, SPIN and GraphCMR can reach the reported results in the respective works. For PARE, the original work trains the model on MPII for pose estimation task and later on EFT-COCO for mesh recovery before training on the full set of datasets. To keep consistent with the practice adopted throughout our work, we benchmark PARE by training it from scratch with only ImageNet initialisation. For Graphormer, the original work evaluates on H36M every epoch before fine-tuning the best H36M model on 3DPW-train (Protocol 1) for 5 epochs. To keep consistent with the evaluation settings throughout this work, we train each model for 100 epochs and report the best PA-MPJPE on 3DPW-test set (Protocol 2). We provide the training logs for all the experiments in `https://github.com/smplbody/hmr-benchmarks`.

3DPW can be generalized to other test sets as well. This is quite different from H36M: models with good performance is not representative of performance on other test sets.

Table 11: **Correlation of performance on test benchmarks**

|  | EFT-COCO | 3DPW | AGORA | EFT-OCH | EFT-LSPET | MI | MuPots-3D | H36M | Average |
|---|---|---|---|---|---|---|---|---|---|
| **EFT-COCO** | 1.000 | 0.860 | 0.891 | 0.910 | 0.820 | 0.643 | 0.595 | 0.387 | 0.729 |
| **3DPW** | 0.860 | 1.000 | 0.768 | 0.761 | 0.779 | 0.704 | 0.396 | 0.506 | 0.682 |
| **AGORA** | 0.891 | 0.768 | 1.000 | 0.793 | 0.624 | 0.626 | 0.696 | 0.183 | 0.654 |
| **EFT-OCH** | 0.910 | 0.761 | 0.793 | 1.000 | 0.750 | 0.449 | 0.424 | 0.378 | 0.638 |
| **EFT-LSPET** | 0.820 | 0.779 | 0.624 | 0.750 | 1.000 | 0.562 | 0.372 | 0.438 | 0.621 |
| **MI** | 0.643 | 0.704 | 0.626 | 0.449 | 0.562 | 1.000 | 0.640 | 0.246 | 0.553 |
| **MuPots-3D** | 0.595 | 0.396 | 0.696 | 0.424 | 0.372 | 0.640 | 1.000 | 0.104 | 0.461 |
| **H36M** | 0.387 | 0.506 | 0.183 | 0.378 | 0.438 | 0.246 | 0.104 | 1.000 | 0.320 |

# 7 Conclusion

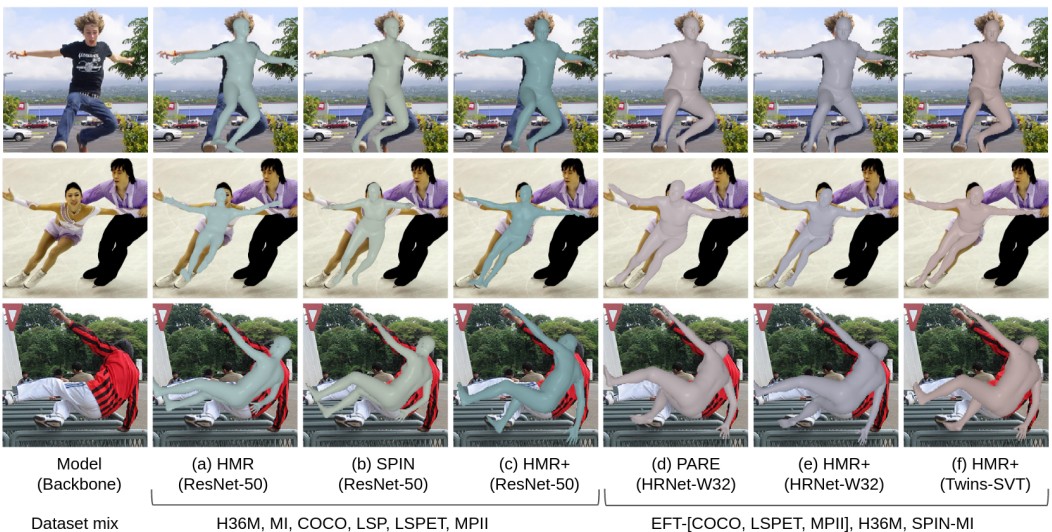

Figure 5: **Qualitative results on COCO and LSPET test sets. From left to right: (a) HMR [30], (b) SPIN [35], (c) HMR+ (ResNet-50) (d) PARE [33] (e) HMR+ (HRNet-W32) (f) HMR+ (Twins-SVT). (a)-(c) follow [31]'s dataset mix while (d)-(f) follow [33]'s dataset mix. HMR+ adopts COCO-weight initialization, L1 loss and selective augmentation. More examples in Appendix F.**

Large amounts of efforts have been devoted to the exploration of novel algorithms for 3D human mesh recovery. However, there are also other important factors that can affect the model performance, which are rarely investigated in a systematic way. To the best of our knowledge, this paper presents the *first* large-scale benchmarking of various configurations for mesh recovery tasks. We identify the key strategies and remarks that can significantly enhance the model performance. We believe this benchmarking study can provide strong baselines for unbiased comparisons in mesh recovery studies. We summarize all our findings in Appendix A.

**Future works.** There are a couple of future research directions. (1) Due to the large amount of experiments, we mainly perform the benchmarks on HMR, which is an important milestone work with straightforward architecture. We provide some evaluation results on a few other algorithms in Section 6 to show the generalization of our major findings. In the future, we plan to extend our studies to more 3D human pose and mesh reconstruction algorithms. (2) Currently we need to use prior knowledge to manually select the datasets and their partitions. Future efforts could investigate if it would be possible to automatically determine the optimal selection of datasets and partitions. For instance, we find that dataset-level weighting is more effective than sample-level weighting. If we consider dataset partition as a hyperparameter to tune, we can borrow techniques from automatic hyperparameter tuning with methods such as reinforcement learning or bayesian optimization to automate dataset configurations. (3) In this paper, we experimentally disclose some inspiring conclusions about HMR training. It is worth conducting deeper investigations to interpret and explain those findings, and obtain the optimal strategy. This will be our future work as well.

## Acknowledgements

We sincerely thank the anonymous reviewers for their valuable comments on this paper. This work is supported by NTU NAP, MOE AcRF Tier 2 (T2EP 20221-0033), and under the RIE2020 Industry Alignment Fund - Industry Collaboration Projects (IAF-ICP) Funding Initiative, as well as cash an in-kind contributions from the industry partner(s).

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
