# A   Lessons from Our Benchmarking

We summarise our findings and open questions raised by these findings:

**Datasets**.

1. The selection of datasets and their relative contributions are important factors to determine the model performance. To fairly evaluate and compare the impact of other factors (e.g., training algorithms), it is crucial to keep the same dataset combination configuration, which is usually ignored by prior works.

2. Diversity of attributes (e.g., human pose and shape, camera characteristics, backbone features) in training datasets are critical for model performance. High diversity (leading to good overlap of test set distributions) can give more satisfactory results. *Using knowledge of the datasets' train-test distributions, merging training datasets that cover a large diversity in attributes could be effective for training. Diversity of these attributes could guide future works could for creating, enhancing or selecting datasets.*

3. To adjust the contribution of different datasets, direct alteration of partitions (and thus increasing the portion of valuable samples) is more effective than keeping the partitions same while reweighting valuable samples. To get a good baseline model, we recommend to adopt more critical datasets and increase their contribution during training. However, current partitions are still manually defined. *Open questions include how to automatically select datasets or adjust the contribution of datasets for training. A possible direction would be to adopt AutoML approaches and consider partitions as hyperparameters to tune.*

4. Addition of SMPL fittings (albeit slightly noisy ones) are still highly effective for training. Meanwhile, noisy keypoints are harmful for model performance. *Addition of pseudo-annotations for existing 2D-keypoint outdoor datasets could be a cost-effective way to enhance existing datasets.*

5. There are also some principles to build robust test sets. Specifically, the test sets should have: (1) accurate ground-truth SMPL annotations captured using mocap or simulation. While EFT-COCO-Val seems like a representative benchmark (i.e. good performance on EFT-COCO-Val correlates to good performance on other benchmarks), we found errors in our visualisation of the SMPL annotations, raising the concern if datasets with pseudo-annotations are suitable to be used as test benchmarks. Currently, 3DPW is the only large-scale real-world outdoor dataset with accurate SMPL ground-truth; (2) large diversity. Diversity in the test set is important and should model closely to real world scenarios. We observe that the widely used test benchmark H36M is not very indicative. Using H36M as the main benchmark would raise concerns if the model is generalisable to a variety of scenarios.

**Backbone and initialization**

1. To fairly evaluate algorithms, it is crucial to properly ablate the backbones and initialisation with conventional ones.

2. Optimal configurations comprise of transformer-based backbones with weights initialized from a strong pose estimation model trained on in-the-wild dataset. *Transferring knowledge from pose estimation models is beneficial for mesh recovery tasks, prompting us to evaluate how we use the same datasets for training.*

**Training strategies**

1. The effect of data augmentations highly depends on the characteristics of the training dataset. Their benefits are more obvious when the training sets contain less diverse and robust samples. When combining multiple datasets during training, we can selectively run data augmentations to different datasets based on their characteristics. *Addition of augmentations could help to enhance existing indoor datasets that lack diversity but contain valuable accurate ground-truth.*

2. Prior works adopt MSE loss for regression of keypoints and SMPL parameters. Using L1 loss instead can not only improve the model's robustness against noisy samples, but also enhance the model performance, especially when the selected datasets are not optimal.

# B  Related Works

For a comprehensive survey on the task of monocular 3D human mesh recovery, Tian et al. [73] has summarized different mesh recovery frameworks and compiled their reported benchmarks. Output types, pseudo labels, datasets and evaluation protocols were factors suggested by Tian et al. [73] that would lead to fluctuations in model performance but no experiments were run to back up their claims. Conversely, our work provided systematic benchmarks and gather insights on how the choice of datasets, architectures and training strategies affect training.

**Datasets** Kanazawa et al. [30] combined Human3.6M (H36M) [23], MPI-INF-3DHP [51], COCO [45], LSPET [27], LSP [26] and MPII [1]. To leverage on multiple datasets, datasets are concatenated according to a manually defined sampling ratio to prevent datasets with a huge amount of samples (i.e. H36M) from overwhelming the model [30, 35]. Recently, more competitive datasets are introduced [29, 6] for training high-performing models.

Table 12: **Summary of the datasets used in various mesh recovery methods and their reported performance (PA-MPJPE in** $mm$**) on 3DPW and H36M datasets**. Abbreviation for the dataset - Human3.6M [23]: H36M, MPI-INF-3DHP [51]: MI, MuCo-3DHP [52]: MuCo, PoseTrack [2]: PT, OCHuman [86]: OCH. 3DPW *Protocol 2* (P2) refers to the evaluation (PA-MPJPE) on 3DPW test set without training on 3DPW train set while *Protocol 1* (P1) includes fine-tuning on 3DPW train set. We use the notation [*]$_{EFT/\ SPIN/\ DP/\ SMPLify-X}$ to denote datasets with EFT, SPIN, DensePose or SMPLify-X fittings.

| Method | Datasets used | Backbones | Losses | 3DPW (P2)↓ | 3DPW (P1)↓ | H36M↓ |
|---|---|---|---|---|---|---|
| HMR [30] | H36M, MI, COCO, LSP, LSPET, MPII | ResNet-50 | Mixed | 76.7 | - | 56.8 |
| NBF [57] | H36M, UP-3D, HumanEva-I | ResNet-50 | Mixed | - | - | 59.9 |
| GraphCMR [36] | H36M, UP-3D, COCO, LSP, MPII | ResNet-50 | Mixed | 70.2 | - | - |
| HoloPose [18] | H36M, MPII, [COCO]$_{DP}$ | ResNet-50 | - | - | - | 46.5 |
| SPIN [35] | H36M, [MI]$_{SPIN}$, COCO, LSP, LSPET, MPII | ResNet-50 | Mixed | 59.2 | - | 41.1 |
| Jiang et al. [24] | H36M, MI, PT, LSP, LSPET, MPII, COCO | ResNet-50 | - | | | 52.7 |
| Zhang et al. [87] | H36M, [COCO]$_{DP}$, UP3D, [LSP, LSPET, MPII, COCO]$_{SPIN}$ | - | - | | | 41.7 |
| Pose2Mesh [8] | MuCo, [H36M]$_{SMPLify-X}$, COCO, Freihand | PoseNet | - | 58.9 | | 47 |
| HKMR [15] | H36M, MI, COCO, LSP, LSPET, MPII | ResNet-50 | L1 | | | 43.2 |
| I2L-MeshNet [53] | MuCo, [H36M]$_{SMPLify-X}$, COCO, Freihand | ResNet-50 | - | 57.7 | | 41.1 |
| DaNet [84] | H36M, [COCO]$_{DP}$, UP3D, [LSP, LSPET, MPII, COCO]$_{SPIN}$ | - | - | 54.8 | | 40.5 |
| Pose2Pose [54] | MuCo, [H36M]$_{SMPLify-X}$, COCO-Wholebody, Freihand | - | - | 55.3 | | 47.4 |
| HybrIK [40] | H36M, MI, COCO | ResNet-34 | - | 48.8 | | |
| METRO [43] | H36M, UP-3D, MuCo, COCO, MPII, Freihand | HRNet-W64 | - | | 47.9 | 36.7 |
| BMP [85] | H36M, MI, MuCo, COCO, LSP, LSPET, PT, MPII | ResNet-50 | MSE | 63.8 | | 51.3 |
| HUND [81] | H36M, 3DPW, COCO-2017, OpenImages | - | - | 57.5 | | 53 |
| EFT [29] | [COCO, MPII, LSPET]$_{EFT}$ | ResNet-50 | Mixed | 54.2 | 52.2 | |
| ProHMR[37] | H36M, [MI, COCO, MPII]$_{SPIN}$ | ResNet-50 | Mixed | | 59.8 | 41.2 |
| DSR [12] | H36M, MI, [COCO]$_{EFT}$ | ResNet-50 | Mixed | 54.1 | 51.7 | |
| ROMP [71] | H36M, UP-3D, [MI, COCO, MPII, LSP]$_{SPIN}$, AICH | ResNet-50 | - | 54.9 | 62 | |
| ROMP[71] | H36M, UP-3D, [MI, COCO, MPII, LSP]$_{SPIN}$, AICH, PT, CrowdPose, MuCo, OH | ResNet-50 | - | 53.3 | 56.8 | |
| Graphormer [44] | H36M, MuCo, UP-3D, COCO, MPII | HRNet-W64 | L1 | | 45.6 | 34.5 |
| THUNDR [82] | H36M, 3DPW, COCO-2017, OpenImages | ResNet-50 | - | 51.5 | | 39.8 |
| PyMAF [83] | H36M, [MI]$_{SPIN}$, COCO, LSP, LSPET, MPII | ResNet-50 | - | 58.9 | 51.2 | 40.5 |
| SPEC [34] | Pano360, SPEC-SYN, SPEC-MTP, 3DPW, MI, H36M, [COCO, MPII, LSPET]$_{EFT}$ | ResNet-50 | - | 53.2 | | |
| PARE [33] | [COCO, MPII, LSPET]$_{EFT}$, MI, H36M | ResNet-50 | Mixed | 52.3 | | |
| PARE [33] | [COCO, MPII, LSPET]$_{EFT}$, MI, H36M | HRNet-W32 | Mixed | 50.9 | 46.5 | |

Table 12 summarises the datasets used in various human mesh recovery algorithms. Many algorithms are trained on their own unique combination of datasets and their best score on 3DPW-test set is directly compared to other methods trained with a different dataset mix.

To complicate matters, Zanfir et al. [81] noted that multiple protocols have also been used for testing. Following SPIN [35], the majority of approaches evaluated on 3DPW test set without any fine-tuning on the training set (protocol 2). However, there are also a number of papers in which 3DPW train set is used during training (protocol 1) [34, 80, 48, 43, 68, 19]. On H36M[23], there are at least 4 protocols: the ones originally proposed by the dataset creators, on the withheld test set of Human3.6M, or protocols 1 and 2 proposed by Kolotouros et al. [35] by re-partitioning the original training and validation sets for which ground truth is available. More recently, Zanfir et al. [81] added evaluation on Panoptic-test [28] and MuPoTs-3D-test [52], Patel et al. [58] recommended AGORA-test and Joo et al. [29] suggested EFT-OCHuman-test and EFT-LSPET-test for more challenging benchmarks.

**Architectures** Following Kanazawa et al. [30], ResNet-50 [21] is the default backbone in many mesh recovery methods [29, 33, 32, 35]. More recently, Kocabas et al. [33] adopted HRNet-W32 [70] in place of ResNet-50 [21] and attributed the performance gains to HRNet-W32's [70] ability to produce more robust high-resolution representations. Cai et al. [5] has also studied other backbone options,

including deeper CNNs such as ResNet-101 and 152 [21], as well as DeiT [74], a vision transformer. Expectedly, larger models demonstrate better capabilities [5], although Cai et al. [5] did not find that vision transformers improve performance over CNN-based ones.

**Training strategy**

A mixture of losses has been typically used in mesh recovery tasks following Kanazawa et al. [30]. Mean Squared Error (MSE) loss is typically used for keypoints supervision, while L1 loss is used for supervision of SMPL parameters.

Various augmentation methods used in pose estimation works have been adopted for mesh recovery [33, 35, 85, 15, 52, 66, 29, 67, 59, 10, 79]. However, varying effectiveness has been reported. Georgakis et al. [15] and Zhang et al. [85] found that occlusion is highly effective while minor performance gains are observed by Kocabas et al. [32]. Joo et al. [29] found that applying extreme crop augmentation only marginally improves performance, while Kocabas et al. [33] reported that cropping harms performance on 3DPW benchmarks. In the above experiments, different dataset mixes and benchmarks are used, warranting the need for a more rigorous investigation into the effect of augmentation on individual datasets.

## C  Occlusion

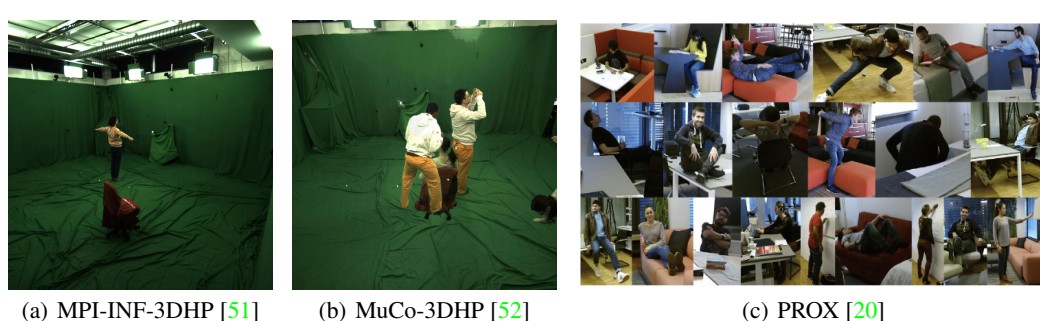

    (a) MPI-INF-3DHP [51]    (b) MuCo-3DHP [52]    (c) PROX [20]

Figure 6: **Example images sourced from (a) MPI-INF-3DHP [51] (b) MuCo-3DHP [52] and (c) PROX [20].**

In fully in-the-wild settings, people often appear under occlusion either due to self-overlapping body parts, close contact with other persons, or interactions with the environment [58]. Person-person or person-object occlusion can be a more important factor that predominates the background. This can be observed from two cases (Fig. 6). (1) MuCo-3DHP [52] is a dataset created through compositing MPI-INF-3DHP [51] with the inter-person occlusion. From Table 2, we observe that the HMR model trained with MuCo-3DHP has better performance than that with MPI-INF-3DHP (78.05 versus 107.15 in PA-MPJPE ($mm$)). (2) PROX [20], despite being the only indoor 2D keypoint dataset, gives the best performance compared with other outdoor 2D keypoint datasets with the PA-MPJPE of 84.69 ($mm$) on 3DPW. PROX contains numerous instances of people interacting with the indoor furniture (Fig. 6), which can improve the model performance.

Whilst keeping other factors constant with the same indoor background, lighting and actors, training with MuCo-3DHP [52] could boast significant improvement gains by adding person-person occlusion (Fig. 6). This is also evident in distributions for pose (see Figure 12), shape (see Figure 13), camera (see Figure 14), and backbone features (see Figure 15) where the distributions of MuCo-3DHP [52] are closer to 3DPW-test [76] and other in-the-wild datasets as compared to MPI-INF-3DHP [51].

## D  Noisy Samples

**Proportion of noisy samples.** We inject noise to different ratios of samples under two situations: (1) only SMPL annotations is noisy (Fig. 7a), which might occur when challenging poses are wrongly fitted; (2) both keypoints and SMPL annotations are noisy (Fig. 7b) as incorrect keypoint estimations lead to erroneous fittings.

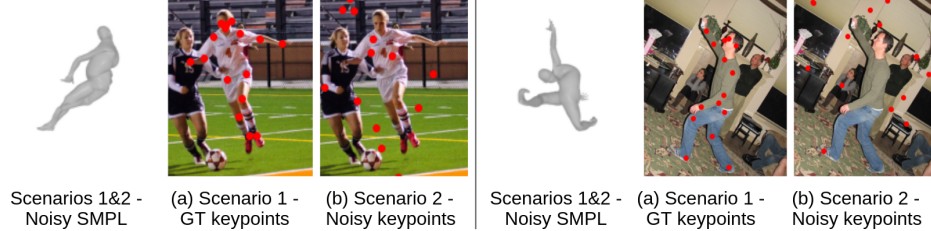

| Scenarios 1&2 - Noisy SMPL | (a) Scenario 1 - GT keypoints | (b) Scenario 2 - Noisy keypoints | Scenarios 1&2 - Noisy SMPL | (a) Scenario 1 - GT keypoints | (b) Scenario 2 - Noisy keypoints |

Figure 7: **Examples of different ratio of noise. On top of noisy SMPL, (a) ground-truth (GT) keypoints are used in scenario 1 while (b) noisy keypoints are used in scenario 2.**

**Scale and location of noise.** Two scenarios were considered for controlled noise about the scale and location. (1) We added noise to all SMPL annotations and vary its scales (i.e., standard deviation). The generated poses are still realistic (Fig. 8a). (2) We observe that fitted poses of certain body parts (i.e. feet and hand) tend to be less accurate in existing fittings. We simulate cases for wrongly fitted body parts by replacing a percentage of pose parameters (body parts) with random noise (Fig. 8b).

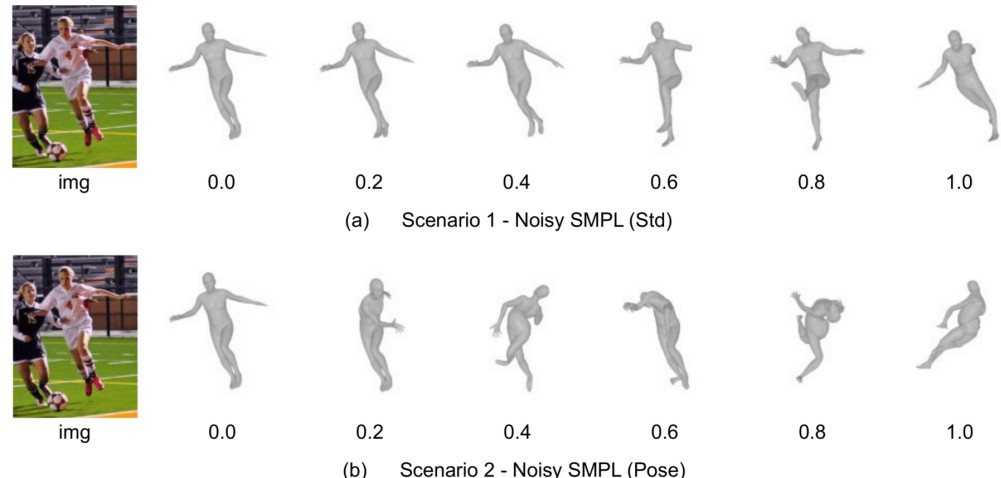

Figure 8: **Examples of different scale and location of noise.**

## E    Augmentation

Fig. 9 compares the training curves without and under different types of augmentation. Training without augmentation increases the indoor-outdoor domain gap, as evidenced from the increasing errors (PA-MPJPE in $mm$) on 3DPW throughout the training episode. Addition of augmentation helps to close the indoor-outdoor domain gap and prevents over-fitting (Fig. 9). Amongst the augmentations, self-mixing seems to be the most helpful for H36M.

Fig. 10 compares the distribution of predicted camera features under different augmentations. For the model that is trained on H36M with self-mixing, the distribution of predicted camera attributes is more similar to that predicted by a well-trained model (Fig. 10a). When training with EFT-COCO, applying augmentation has a minute effect on the camera distribution (Fig. 10b), probably due to the diverse variety of camera angles present. This could explain why applying augmentation to EFT-COCO has a less pronounced effect on 3DPW performance (Table 8).

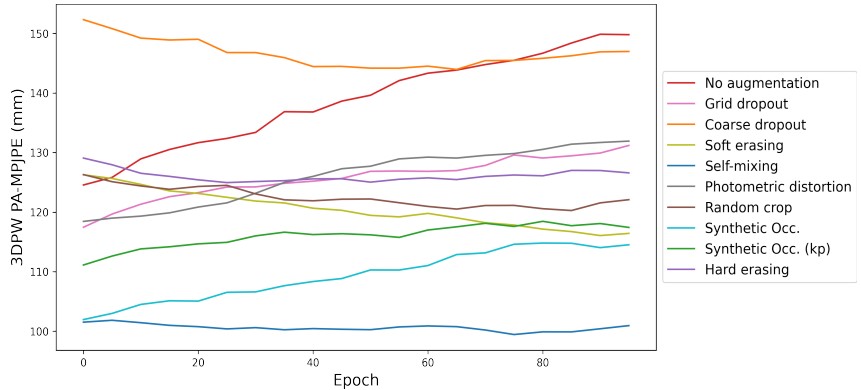

Figure 9: **Per-epoch evaluation on 3DPW (PA-MPJPE in** $mm$ **when trained on H36M under different augmentations.**

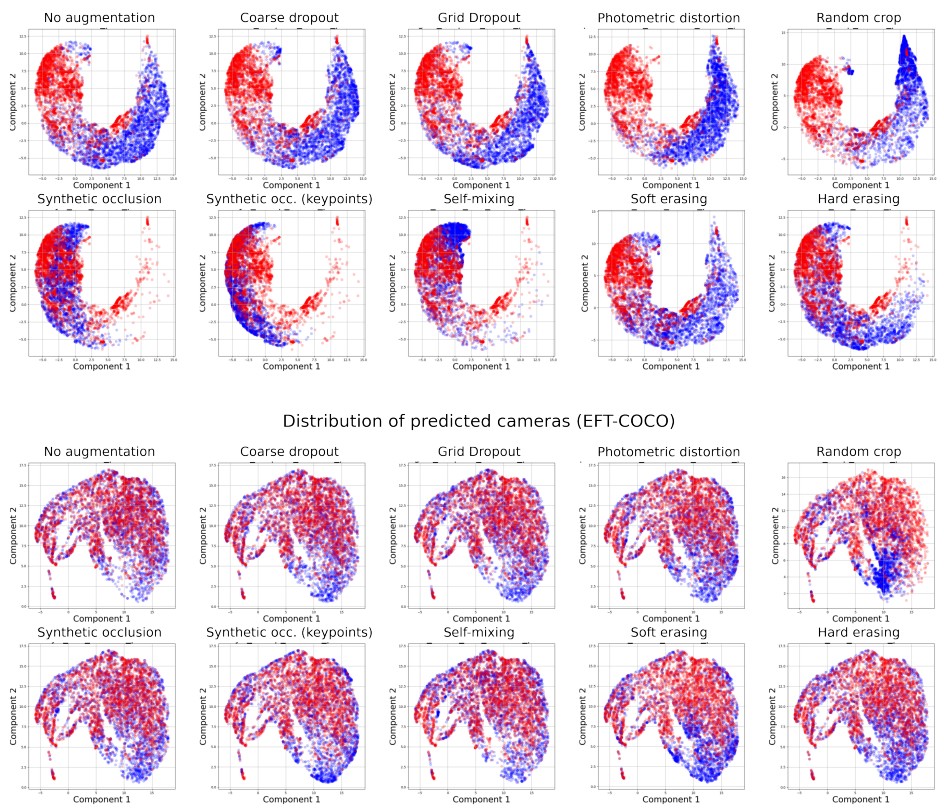

Figure 10: **Effect of applying augmentation on the distribution of predicted camera features for (top) H36M and (bottom) EFT-COCO.**

# F    Qualitative evaluation

Under the same model capacity and dataset mixes, our variant (HMR+) outperforms HMR [30] and SPIN [35] both qualitatively (Fig. 11) and quantitatively (Table 1). HMR+ adopts the training strategies of COCO-weight initialization, L1 loss and selective augmentation. Using the same dataset selection and backbone (HRNet-W32) as PARE [33], qualitative and quantitative differences are more subtle as PARE [33] is already a robust model (Fig. 11).

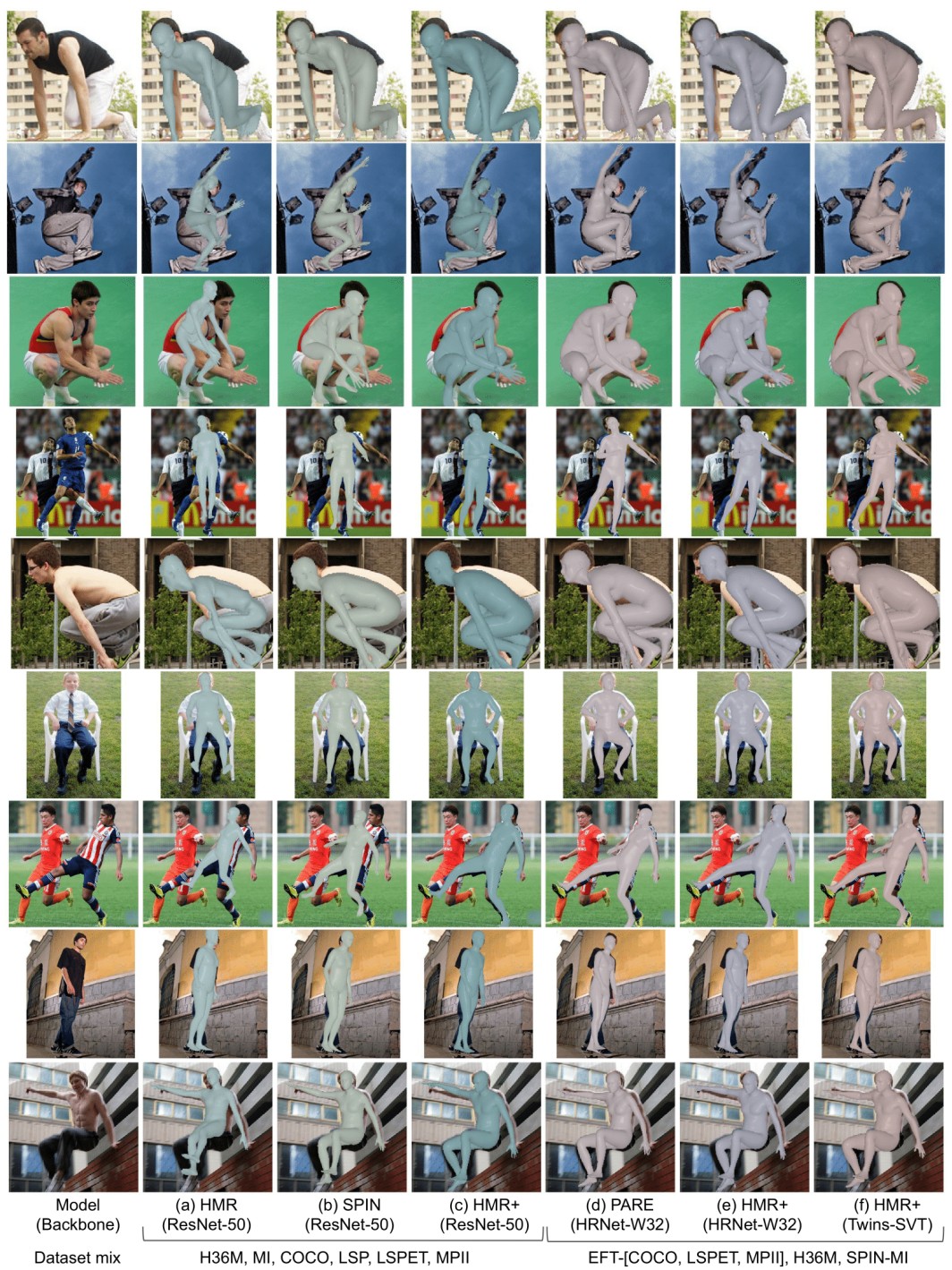

| Model (Backbone) | (a) HMR (ResNet-50) | (b) SPIN (ResNet-50) | (c) HMR+ (ResNet-50) | (d) PARE (HRNet-W32) | (e) HMR+ (HRNet-W32) | (f) HMR+ (Twins-SVT) |
|---|---|---|---|---|---|---|
| Dataset mix | | H36M, MI, COCO, LSP, LSPET, MPII | | | EFT-[COCO, LSPET, MPII], H36M, SPIN-MI | | |

Figure 11: **Qualitative results on COCO, LSPET and OCHuman test sets. From left to right: (a) HMR [30], (b) SPIN [35], (c) HMR+ (Ours) with ResNet-50 backbone (d) PARE [33] (e) HMR+ (Ours) with HRNet-W32 backbone (f) HMR+ (Ours) with Twins-SVT backbone. (a)-(c) follow [31]'s dataset mix while (d)-(f) follow [33]'s dataset mix. HMR+ adopts COCO-weight initialization, L1 loss and selective augmentation.**

# G   Other benchmarks

Table 13: **HMR model performance (PA-MPJPE in** $mm$**) on the 3DPW [76] and H36M[23] test sets when trained on individual datasets. For PROX and MuPoTs-3D, only 2D keypoints are used for training. P: person-person occlusion O: person-object occlusion.**

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

Table 14: **HMR model performance (PA-MPJPE in** $mm$**) on 3DPW with different backbone architectures.**

| Backbone | Params (M) | FLOPs (G) | 3DPW↓ | H36M↓ |
|---|---|---|---|---|
| ResNet-50 [21] | 28.79 | 4.13 | 64.55 | 46.47 |
| ResNet-101 [21] | 47.78 | 7.83 | 63.36 | 47.50 |
| ResNet-152 [21] | 63.42 | 11.54 | 62.13 | 47.33 |
| HRNet [70] | 36.69 | 11.05 | 64.27 | 49.95 |
| EfficientNet-B5 [72] | 33.62 | 0.03 | 65.16 | 44.31 |
| ResNext-101 | 91.39 | 16.45 | 64.95 | 50.76 |
| Swin [46] | 51.72 | 32.48 | 62.78 | 46.79 |
| ViT [11] | 91.07 | 11.29 | 62.81 | 49.06 |
| Twin-SVT [9] | 59.27 | 8.35 | 60.11 | 46.08 |
| Twin-PCVCT [9] | 47.02 | 6.45 | **59.13** | 48.16 |

Table 15: **HMR model performance (PA-MPJPE in** $mm$**) when trained with different combinations of datasets.**

| Mix | Datasets | 3DPW↓ | H36M↓ |
|---|---|---|---|
| 1 | H36M, MI, COCO | 66.14 | 48.90 |
| 2 | H36M, MI, EFT-COCO | 55.98 | 45.18 |
| 3 | H36M, MI, EFT-COCO, MPII | 56.39 | 46.06 |
| 4 | H36M, MuCo, EFT-COCO | 53.90 | 46.01 |
| 5 | H36M, MI, COCO, LSP, LSPET, MPII | 64.55 | 49.47 |
| 6 | EFT-[COCO, MPII, LSPET], SPIN-MI, H36M | 55.47 | 46.44 |
| 7 | EFT-[COCO, MPII, LSPET], MuCo, H36M, PROX | 52.96 | 51.20 |
| 8 | EFT-[COCO, PT, LSPET], MI, H36M | 55.97 | 46.14 |
| 9 | EFT-[COCO, PT, LSPET, OCH], MI, H36M | 55.59 | 47.35 |
| 10 | PROX, MuCo, EFT-[COCO, PT, LSPET, OCH], UP-3D, MTP, Crowdpose | 57.80 | 50.51 |
| 11 | EFT-[COCO, MPII, LSPET], MuCo, H36M | 52.54 | 47.19 |

## H    Optimized configurations for other algorithms

In addition to Table 10, Table 16 considers different dataset mixes and backbones for the additional algorithms we included. Similar to HMR, high-quality models for other algorithms are also established with optimized dataset mixes, backbones and training strategies.

Table 16: **Model performance of other algorithms with optimized configurations on the 3DPW test set.**
Abbreviations for the datasets - Human3.6M [23]: H36M, MPI-INF-3DHP [51]: MI, MuCo-3DHP [52]: MuCo

| Algorithm | Dataset | Backbone | Variant | PA-MPJPE↓ | MPJPE↓ | PA-PVE↓ | PVE↓ |
|---|---|---|---|---|---|---|---|
| PARE [33] | EFT-[COCO, LSPET, MPII], H36M, SPIN-MI | HrNet-W32 | EFT-COCO | 50.90 | 82.0 | - | 97.9 |
| PARE (Ours) | EFT-[COCO, LSPET, MPII], H36M, SPIN-MI | HrNet-W32 | - | 61.99 | 109.82 | 82.33 | 133.86 |
| PARE (Ours) | EFT-[COCO, LSPET, MPII], H36M, SPIN-MI | HrNet-W32 | L1-COCO-Aug | 58.32 | 100.35 | 77.22 | 121.97 |
| PARE (Ours) | EFT-[COCO, LSPET, MPII], H36M, SPIN-MI | Twins-SVT | L1-COCO-Aug | 51.96 | 93.46 | 81.33 | 130.20 |
| PARE (Ours) | EFT-[COCO, LSPET, MPII], H36M, MuCo | Twins-SVT | L1-COCO-Aug | 51.93 | 91.43 | 68.40 | 110.32 |
| GraphCMR [36] | COCO, H36M, MPII, LSPET, LSP, UP3D | ResNet-50 | - | 70.52 | 116.83 | 87.50 | 133.67 |
| GraphCMR | COCO, H36M, MPII, LSPET, LSP, UP3D | ResNet-50 | L1-COCO-Aug | 60.26 | 99.28 | 75.75 | 113.17 |
| GraphCMR | EFT-[COCO, LSPET, MPII], H36M, SPIN-MI | ResNet-50 | - | 60.51 | 101.69 | 77.51 | 121.37 |
| GraphCMR | EFT-[COCO, LSPET, MPII], H36M, SPIN-MI | Twins-SVT | L1-COCO-Aug | 53.29 | 91.07 | 70.52 | 108.14 |
| SPIN [35] | H36M, MI, COCO, LSP, LSPET, MPII | ResNet-50 | - | 59.2 | 96.9 | - | 135.1 |
| SPIN (Ours) | H36M, MI, COCO, LSP, LSPET, MPII | ResNet-50 | L1-COCO-Aug | 50.54 | 80.49 | 68.29 | 96.67 |
| SPIN (Ours) | EFT-[COCO, LSPET, MPII], H36M, SPIN-MI | ResNet-50 | L1-COCO-Aug | 55.28 | 93.52 | 72.19 | 109.57 |
| SPIN (Ours) | EFT-[COCO, LSPET, MPII], H36M, SPIN-MI | HRNet-W32 | L1-COCO-Aug | 47.59 | 80.77 | 64.22 | 96.22 |
| MeshGraphormer [44] | H36M, COCO-2017, UP3D, MPII, MuCo | HRNet-W48 | - | 63.18 | 108.02 | 76.05 | 125.56 |
| MeshGraphormer (Ours) | H36M, COCO-2017, UP3D, MPII, MuCo | HRNet-W48 | L1-COCO-Aug | 58.82 | 104.63 | 76.79 | 132.52 |
| MeshGraphormer (Ours) | H36M, COCO-2017, UP3D, MPII, MuCo | Twins-SVT | L1-COCO-Aug | 58.13 | 98.03 | 73.32 | 116.95 |
| MeshGraphormer (Ours) | H36M, COCO-2017, UP3D, EFT-MPII, MuCo | Twins-SVT | L1-COCO-Aug | 58.30 | 96.71 | 74.88 | 124.97 |

## I    Feature distributions of datasets

As several datasets do not contain ground-truth camera angles and poses, we trained a robust HMR (3DPW errors of 51.66 $mm$) to obtain estimations of four attributes: 1) pose $\theta \in R^{69}$ modeled by relative 3D rotation of K = 23 joints in axis-angle representation, 2) shape $\beta \in R^{10}$ parameterized by the first 10 coefficients of a PCA shape space, 3) camera translation $t^c \in R^3$ obtained by predicting weak perspective camera parameters, and 4) features $f \in R^{2048}$ obtained from the ResNet-50 backbone.

Following which, the distribution of each attribute is visualized after dimension reduction with Uniform Manifold Approximation and Projection (UMAP) [49]. For visualization purposes, we randomly downsample data points from each dataset and compare them with the features reduced from 3DPW-test set.

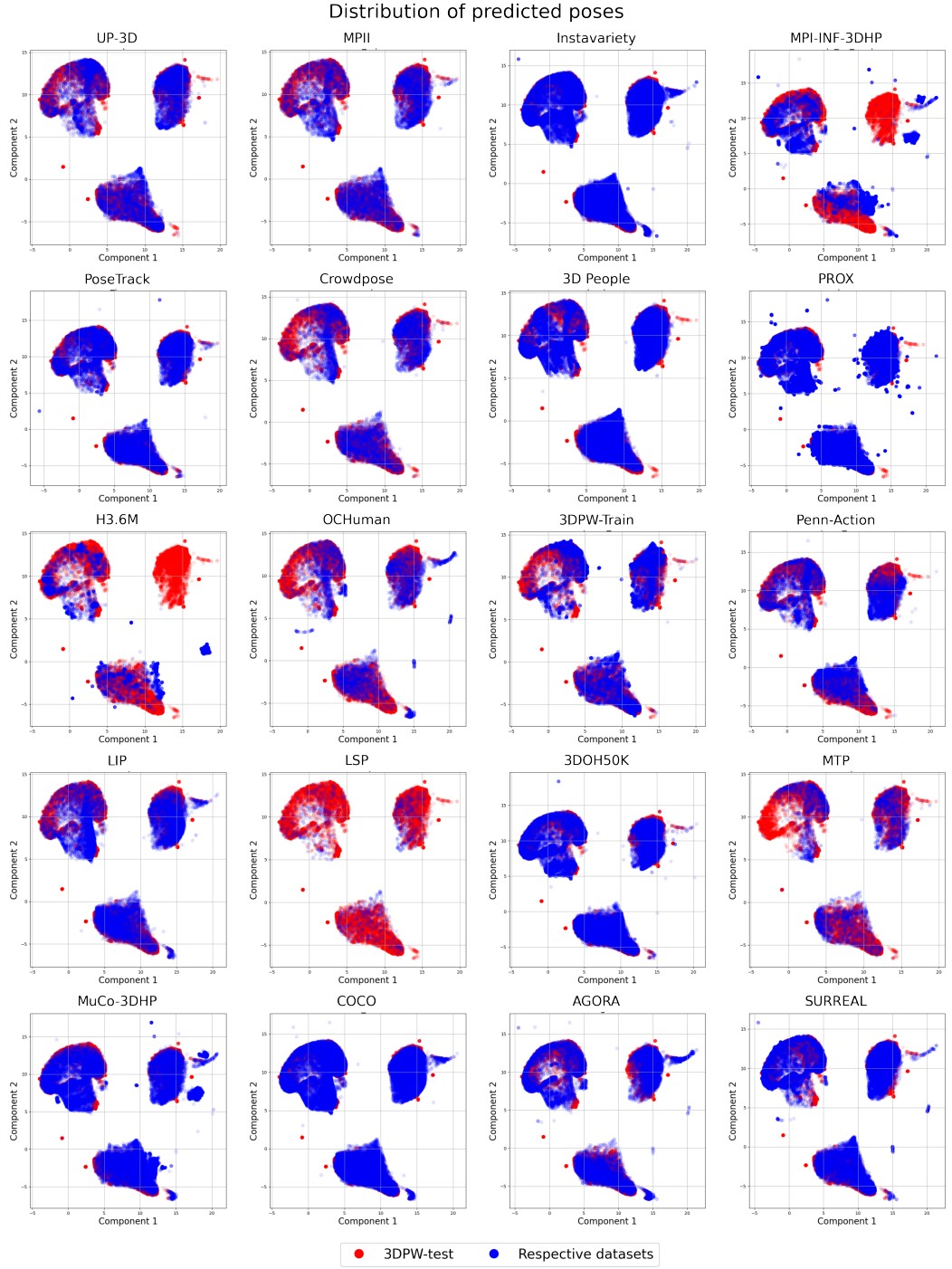

Figure 12: **Feature distribution of poses between 3DPW-test (red) and the respective datasets (blue).**

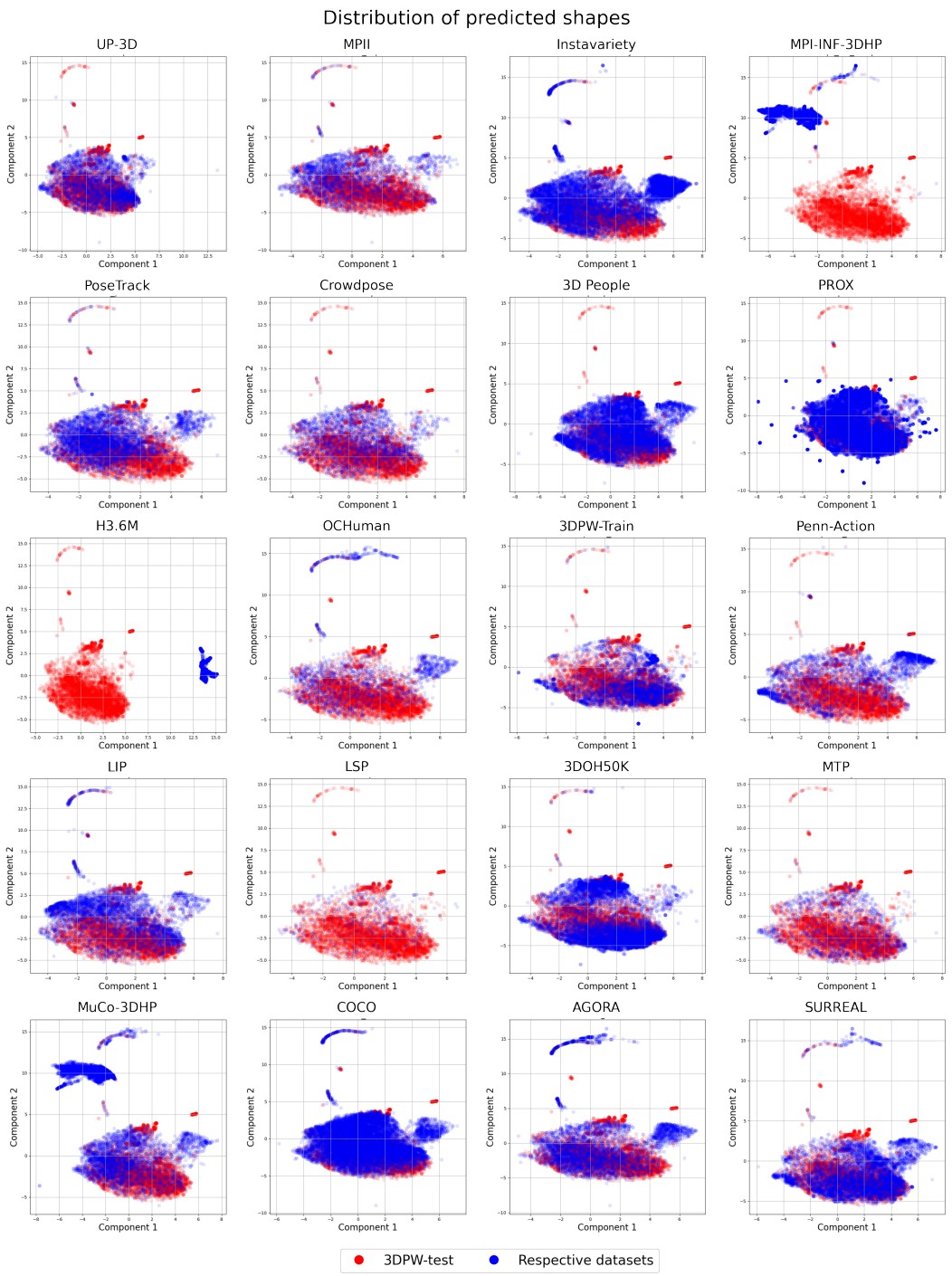

Figure 13: **Feature distribution of shapes between 3DPW-test (red) and the respective datasets (blue). Notably, datasets such as Instavariety [31], PROX [20], COCO [45], AGORA [58] contain a diverse range of shapes. Meanwhile, indoor datasets such as MPI-INF-3DHP [51] and H36M [23] have a rather distinct distribution from 3DPW-test, which could be attributed to the small number of subjects in each dataset. MuCo-3DHP [52] is the variant of MPI-INF-3DHP [51] that contains person-person occlusion. This helps to increase diversity and close the distribution shift between 3DPW-test.**

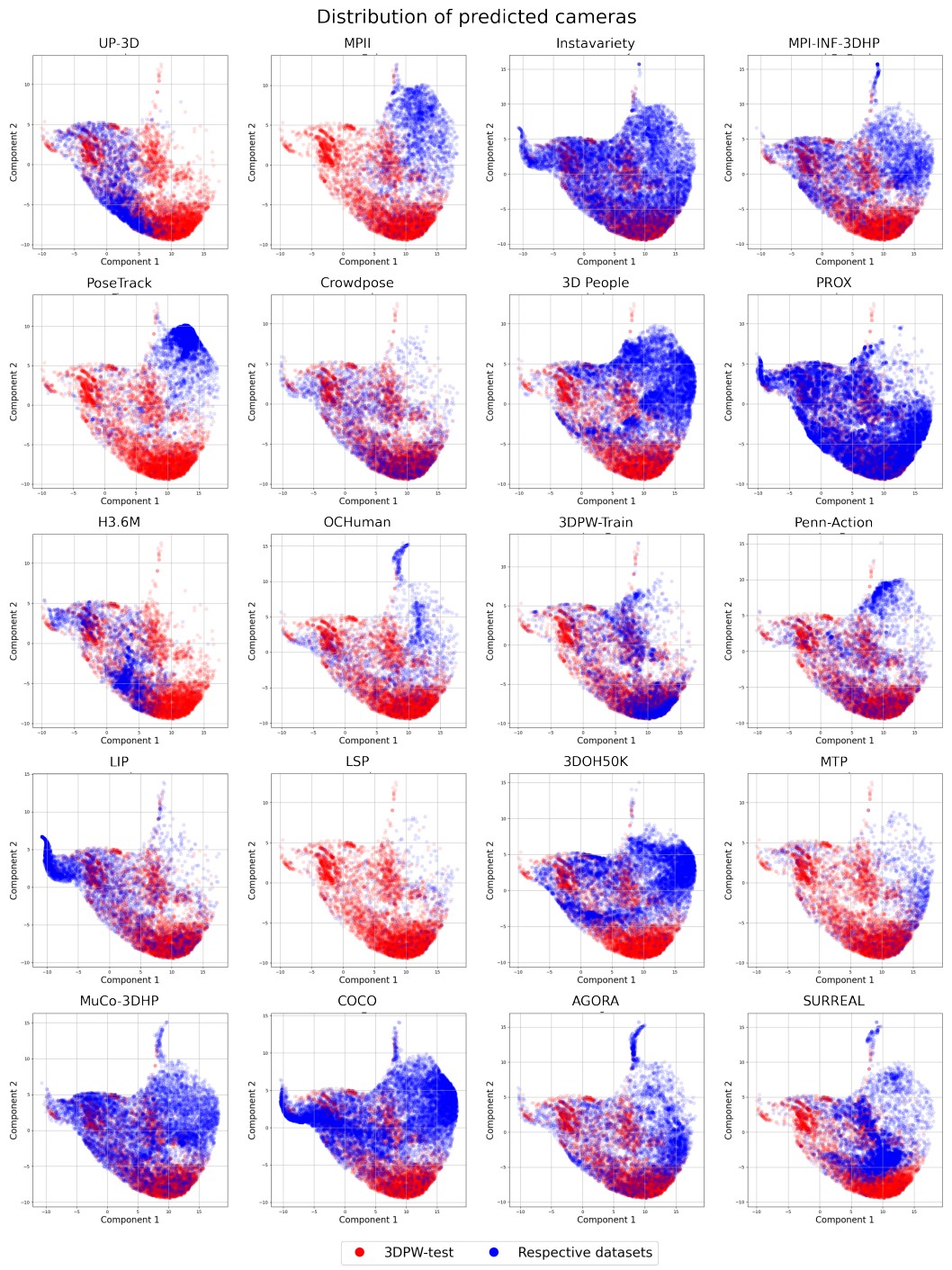

Figure 14: **Feature distribution of estimated cameras between 3DPW-test (red) and the respective datasets (blue). Amongst datasets with only 2D keypoints, Instavariety [31], PROX [20] and COCO [45] have a more diverse distribution, as compared to MPII, PoseTrack, OCHuman, LIP or Penn-Action. This might also explain they achieve more competitive results on 3DPW-test benchmarks.**

Distribution of predicted features

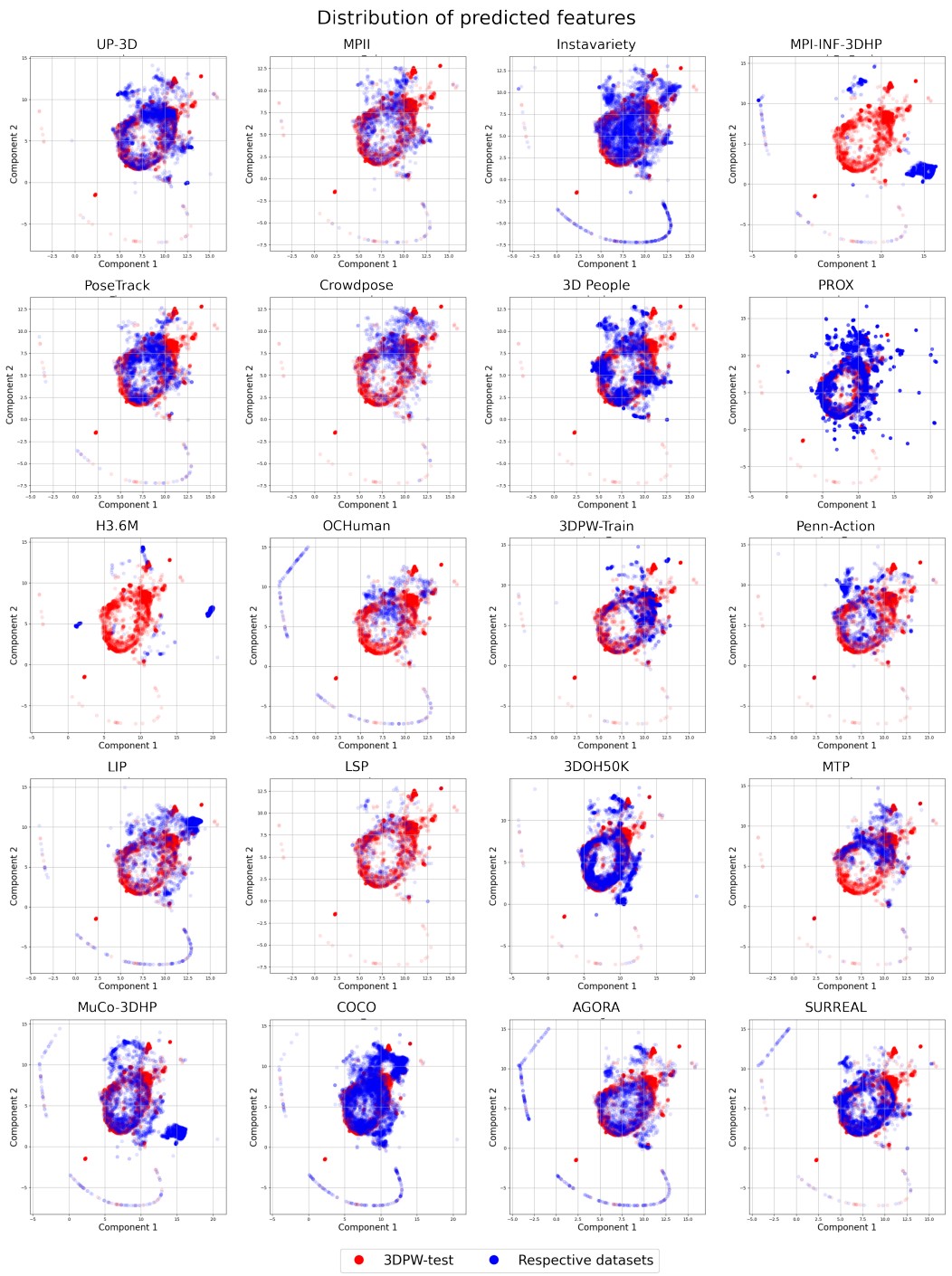

Figure 15: **Feature distribution of backbone features between 3DPW-test (red) and the respective datasets (blue). Notably, Instavariety, COCO contain a diverse range of backbone features. Meanwhile, indoor datasets such as MPI-INF-3DHP [51] and H36M [23] have a rather distinct distribution from 3DPW-test, which could be attributed to the same colored background in both datasets. MuCo-3DHP [52] is the variant of MPI-INF-3DHP [51] that contains augmented backgrounds. This helps to increase diversity and close the distribution shift between 3DPW-test.**