# OpenReview forum: "Benchmarking and Analyzing 3D Human Pose and Shape Estimation Beyond Algorithms"
_NeurIPS.cc/2022/Track/Datasets_and_Benchmarks — NeurIPS 2022 Datasets and Benchmarks _

### Official Review · Reviewer_H6N5 · 2022-07-24
**A solid contribution to 3D human recovering**

**Rating:** 7
**Confidence:** 3
**Correctness:** Yes, easy to follow.
**Clarity:** Good.

**Strengths:**

+ The discussions and experiments are inspiring and make solid contributions.

+ The results of the large-scale experiments reveal several important discoveries, e.g., the mixed training of multiple datasets, the differences between various backbones, whether the noisy reconstructions affect the training, etc.

+ Thorough details and settings of the experiments are given in the suppl and main text, which would help the readers to understand this work better.

+ The project page is organized well with detailed documentation, open models, and implementations which are very useful for the community.

**Weaknesses:**

- Contributions to dataset/benchmark: though I think this is a solid work, the last two contributions may be kind of off the point of the NeurIPS dataset and benchmark track. The backbone and training strategies analysis are more like the method contribution. But I think this is not a big deal considering the solid works.

- Lacking possible advice on how to build a better test set for this direction. Though the discussed factors are important, the relations between the train and test data also matter a lot. Existing test sets also have many problems and possible aspects to improve. Please give some discussions about this point which would make this work more solid and inspiring.

- There is no comparison between different methods given the same set of training datasets to clearly indicate how they perform in a fair comparison. If possible, I think this would make the first contribution better and more beneficial.

**Additional Feedback:**

1. In some tables like Tab. 1, 2, and 4, maybe some bold scores can make the comparison clear like Tab. 6.

2. L165, advanced algorithm. I suggest changing this to a more sophisticated algorithm or other adjectives as no free lunch.

3. I like the augmentation analysis. But the two sets of results give the opposite conclusion. Though I agree that different datasets have different properties thus will give different reactions to the augmentations. However, could the authors give some more instructive advice for future works?

**Documentation:**

A good project page with detailed documentation has been provided.

**Ethics:**

No concerns.

**Relation To Prior Work:**

Yes, and sound.

**Summary And Contributions:**

This paper gives a detailed analysis of 3D human recovery in datasets, backbones, and training policies. Inspiring and detailed discussions and sound-designed experiments are conducted. Besides, a project page with detailed documentation is also provided which would help the community for future work. Overall, I think this is a solid and good work for 3D human-related works.

---

> ### Author Response · Authors · 2022-08-18
> **Response (1/2)**
>
> We sincerely thank the reviewer for your insightful comments and recognition to this work, especially for acknowledging that inspiring and detailed discussion and sound-designed experiments are conducted. Below we would like to provide point-to-point responses to address all the raised questions:
>
> > Q1: Contributions to dataset/benchmark: though I think this is a solid work, the last two contributions may be kind of off the point of the NeurIPS dataset and benchmark track. The backbone and training strategies analysis are more like the method contribution. But I think this is not a big deal considering the solid works
>
> **A1:** We understand your concern. We believe that our paper can fit for the D&B track for the following reasons.
>
> First, according to the blog post referenced in the FAQs of the D&B track [https://neuripsconf.medium.com/announcing-the-neurips-2021-datasets-and-benchmarks-track-644e27c1e66c],  the scope of this track includes **“other benchmarking efforts to connect models to real world impacts”** – We identified optimal configurations (including backbone and training strategies) that help model training. Second, there is precedence of similar works from NeurIPS’21 D&B track that are similar to our paper. For instance, some papers [1, 2, 3] provide analysis on non-data aspects, i.e., architectures or how to build a more robust model. Third, backbone architectures and training strategies are highly relevant to dataset benchmarking, as the selection of those strategies can highly affect the selection of training datasets, as we discovered in this paper.
>
>
> > Q2: Lacking possible advice on how to build a better test set for this direction. Though the discussed factors are important, the relations between the train and test data also matter a lot. Existing test sets also have many problems and possible aspects to improve. Please give some discussions about this point which would make this work more solid and inspiring
>
> A2: Thanks for the suggestion! In the revision, we have added more experiments about evaluations on 7 other test sets (H36M, AGORA validation, MPI-INF-3DHP test, EFT-COCO validation, MuPots3D-test, EFT-OCHuman test, EFT-LSPET test). We find that there is strong correlation between performance on 3DPW with performance on other test sets, indicating that 3DPW is a fairly good benchmark and robust test set. Please check Section 6 (Table 11, marked in blue) for more details and analysis.
>
> Although 3DPW is a strong indicator, it is not perfect and newer algorithms are reaching saturation on the 3DPW-test set. It can be improved from having more challenging scenarios, poses and greater diversity in actors.
>
> We also discuss the possible factors of building a more robust test set in Appendix A (marked in blue). Specifically, they should have:
> - **accurate ground-truth SMPL annotations** captured using mocap/ simulation. While EFT-COCO-Validation seems like a representative benchmark (i.e. good performance on EFT-COCO-Val correlates to good performance on other benchmarks), we found errors in our visualisation of the SMPL annotations, raising the concern if datasets with pseudo-annotations are suitable to be used as test benchmarks. Currently, 3DPW is the only large-scale real-world outdoor dataset with accurate SMPL ground-truth.
> - **diversity.** Diversity in the test set is important and should model closely to real world scenarios. It is ideal to have diversity in certain attributes we identified, these include camera characteristics (avoid fixed cameras or collect from a variety of angles and heights), poses (more scenarios and actions), shape (include a variety of subjects of different body shapes) and backbone features (a variety of scenes). We observe that the widely used test benchmark H36M is not very indicative. Using H36M as the main benchmark would raise concerns if the model is generalisable to a variety of scenarios.
>
> > Q3: There is no comparison between different methods given the same set of training datasets to clearly indicate how they perform in a fair comparison. If possible, I think this would make the first contribution better and more beneficial.
>
> A3: Thank you for the advice. In the revision, we have added new experiments evaluating different network models (SPIN [4], GraphCMR [5], PARE [6], Graphormer [7]). Please check Section 6 for more details and analysis.
>
> > Q4: In some tables like Tab. 1, 2, and 4, maybe some bold scores can make the comparison clear like Tab. 6.
>
> A4: Thank you for the helpful suggestion. We have changed it in Tables 1, 2 and 4 in the revised paper (highlighted in blue).
>
> > Q5: L165, advanced algorithm. I suggest changing this to a more sophisticated algorithm or other adjectives as no free lunch.
>
> A5: Thank you for pointing this out. We have changed it to “sophisticated” algorithm in Section 3.2 of the revised paper (highlighted in blue).

---

> > ### Author Response · Authors · 2022-08-18
> > **Response (2/2)**
> >
> > > Q6: I like the augmentation analysis. But the two sets of results give the opposite conclusion. Though I agree that different datasets have different properties thus will give different reactions to the augmentations. However, could the authors give some more instructive advice for future works?
> >
> > **A6:** Thank you for your encouraging comment. The contradicting conclusions could be explained by the underlying distribution between the training and testing sets (Fig 9 in Appendix). Adding augmentation to H36M (a dataset with limited background subjects) could help to increase diversity and lead to a closer distribution to the test-set. On the other hand, adding augmentation to EFT-COCO causes distribution shift (seen from the visualisation a misalignment in the overlap of parameters between augmented EFT-COCO and 3DPW test set).
> >
> > 3D pose and shape estimation is a rather unique problem where it is standard protocol to train with a mixture of datasets together. To achieve good performance on a specific test set, perhaps one could consider looking at the distributions of individual training sets and the test set before applying selective augmentation for some of them. Directly applying augmentation might bring more harm than good, as EFT [8] reported marginal improvement and PARE[6] reported a worsening in performance when applying crop augmentation. We believe it might have to do with distribution shifts in some of the training datasets. We add more discussions regarding this as future work in Appendix A of the revised paper (marked in blue).
> >
> > References:
> >
> > [1] Chen et al. “Benchmarks for Corruption Invariant Person Re-identification” NeurIPS 2021 Datasets and Benchmarks Track.
> >
> > [2] Li et al. "MQBench: Towards Reproducible and Deployable Model Quantization Benchmark." NeurIPS 2021 Datasets and Benchmarks Track.
> >
> > [3] Yi et al. “Benchmarking the Robustness of Spatial-Temporal Models Against Corruptions.” NeurIPS 2021 Datasets and Benchmarks Track.
> >
> > [4] Kolotouros et al. “Learning to Reconstruct 3D Human Pose and Shape via Model-fitting in the Loop.” ICCV 2019.
> >
> > [5] Kolotouros et al. “Convolutional Mesh Regression for Single-Image Human Shape Reconstruction”. CVPR 2019.
> >
> > [6] Kocabas et al. “PARE: Part Attention Regressor for 3D Human Body Estimation.” ICCV 2021.
> >
> > [7] Lin et al. “Mesh Graphormer”. ICCV 2021.
> >
> > [8] Joo et al. “Exemplar Fine-Tuning for 3D Human Pose Fitting Towards In-the-Wild 3D Human Pose Estimation.” 3DV 2021.
> >
> > Please don’t hesitate to let us know if there are any additional clarifications or experiments that we can offer!

---

### Official Review · Reviewer_4jJK · 2022-07-25
**A good benchmark and anaylsis paper.  Most findings are not surprising but good to have them all in one place for the pose and shape estimation domain.**

**Rating:** 7
**Confidence:** 4

**Strengths:**

This paper makes several overall contributions on several axes. Here is a very brief attempt at outlining the takeaways though it is not necessarily comprehensive.

* Dataset:
    * Diversity in various attributes is useful
    * Having occlusion and SMPL fittings in the ground truth is useful
    * More datasets is not better, it can even be harmful
    * Careful configuration of different contributions (weightings) for different datasets can bring a large improvement.
    * Directly altering the partitions is more effective than reweighting the samples.
* Noise:
    * Some noise in SMPL is ok
    * Noise in keypoints is bad
    * Replacing a percentage of pose parameters (i.e., for some body parts) with random noise (not just adding some noise to good values) is also bad.
* Architecture:
    * Increased capacity in CNNs helps (at a power cost of course)
    * Transformers are better than CNNs at comparable FLOPs
* Training:
    * Directly supervising to SMPL parameters is better than supervising to keypoints or other features.
    * It is helpful to pre-train backbone for a mesh estimation model with a good pose estimation model.
    * Most augmentations help, self-mixing being one of the most useful.
    * Datasets with a lot of diversity don’t need as much augmentation (or it can be harmful), notably when it comes to self-occlusion.
    * L1 loss is better than L2 loss



**Weaknesses:**

Most of the findings are not surprising to a practitioner in the field, although the specific details for the specific datasets and architectures are still quite good to have.  It might be useful to have a summary table at the beginning or end as spelled out in the “Results” blocks throughout the paper.  It would also be good enumerate more of the resulting open questions raised by these findings, especially those that are somewhat surprising (or at least, not obvious).  For instance, why is directly altering the partitions more effective than reweighting the samples?  Why is L1 loss better than L2?  Some explanations are occasionally offered, so the main ask is to offer explanations more consistently.


**Additional Feedback:**

The paper could try to provide more explanations for why some choices are better, or at least state these as outstanding questions for what would help advance the field.  For instance, a main finding is that adding more datasets is not better, and can even be harmful; moreover careful configuration of different contributions (weightings) for different datasets can bring a large improvement.  Of course it’s good to see the specific details about how specific combinations of public datasets impacts algorithms. Yet the paper didn’t link these findings back to the specific diversities in these datasets discussed in the previous section.  One has to do some work to see the contributions of different datasets *by name* and then map that to the contributions of different amounts of *attributes* of these datasets. Even with this, keeping the unit of analysis as an entire dataset is a bit cumbersome. Ideally (hopefully this will come in future work) we would like these findings to lead to a recipe for constructing a training set with ideal combinations of characteristics (whether through merging parts of existing datasets or creating a new one). A similar desire applies to other findings as mentioned above.


**Clarity:**

Quite clear. The “Remarks” blocks are helpful.  Might be useful to put these into a single summary somewhere.


**Correctness:**

Did not check, but the methodology seems sound so I believe it is correct.


**Documentation:**

Does not provide a dataset, so none needed.


**Ethics:**

No ethics concerns.


**Relation To Prior Work:**

Does a good comparison to prior work, indeed the goal is to put more prior work on the same playing-field.


**Summary And Contributions:**

This paper systematically varies several dataset, architecture, and training choices while keeping evaluations consistent to provide a well justified set of best practices, along with specific recommendations when keeping other factors equal.  While many of the general findings are unsurprising (the mixture of datasets can make a big difference, it is useful for datasets to have more diversity), some of the findings aren’t obvious a priori so the evidence provided here is useful (e.g., models tend to be more robust to noise in SMPL parameters than 3D points). Moreover, it is quite useful to have the specific details of datasets and choices in one place, even those that are not surprising. The paper is correct that, since each publication makes different choices in all of the dimensions, it is difficult to make good evidence-based decisions, so this paper makes a significant contribution in reducing some of the ambiguities when datasets, architectures, and training choices aren’t matched.

---

> ### Author Response · Authors · 2022-08-18
> **Response**
>
> We sincerely thank the reviewer for your insightful comments and recognitions to this work, especially for acknowledging that this work makes a significant contribution in reducing some of the ambiguities when datasets, architectures and training choices aren’t matched. Below we would like to provide point-to-point responses to address all the raised questions:
>
> > Q1: Most of the findings are not surprising to a practitioner in the field, although the specific details for the specific datasets and architectures are still quite good to have. It might be useful to have a summary table at the beginning or end as spelled out in the “Results” blocks throughout the paper.
>
> A1: Thank you for your advice! We have added an new section (Appendix A, marked in blue) to summarise the findings and enumerate the recommendations and open-ended questions raised by these findings. We believe these will be very helpful to the researchers and practitioners in this direction.
>
> > Q2: It would also be good enumerate more of the resulting open questions raised by these findings, especially those that are somewhat surprising (or at least, not obvious). For instance, why is directly altering the partitions more effective than reweighting the samples? Why is L1 loss better than L2? Some explanations are occasionally offered, so the main ask is to offer explanations more consistently.
>
> A2: Thank you a lot for your suggestion. It is indeed important to provide more interpretation and explanations about the discoveries in this paper. We have added more explanations in Section 3.2 and discussions in Section 7 of the revised paper (marked in blue). This is a big and challenging topic. Due to the page and time limit, we will seriously consider this as an important future work.
>
> Please don’t hesitate to let us know if there are any additional clarifications or experiments that we can offer!

---

> ### Author Response · Authors · 2022-08-25
> **Follow-up**
>
> Dear reviewer,
>
> We would like to follow up to check if your concerns have been addressed. In the previous response, we have made the following updates/clarification:
> - Regarding your concern on the restructuring of the paper (Q1), we have added a new section (Appendix A) to summarise the findings, recommendations and open-ended questions.
> - Regarding your precious advice to provide more consistent explanations (Q2), we have added linkages in Sections 3.2 and 7.
>
> We are happy to answer further questions.

---

### Official Review · Reviewer_SarV · 2022-07-25
**Interesting insights and best practises into overlook aspects of 3D human pose and shape estimation**

**Rating:** 6
**Confidence:** 4
**Correctness:** I do not have any concerns about the …
**Clarity:** The paper is well written.

**Strengths:**

-Most of the focus in 3D mesh recovery is on algorithmic development with less effort on establishing proper evaluation protocols. Authors demonstrate how commonly overlooked factors like dataset mixing, augmentation strategy and model initialisation can effect reported performance leading to unfair comparison between different methods.

-The investigation performed in this work can serve as a valuable set of best practises for developing mesh recovery models. Multiple factors are explored like optimal dataset mix, value of pretraining, best augmentation strategies and most suitable losses, e.t.c.

-Various interesting insights are raised. One interesting example is that difference in performance cannot always be explained by the indoor-outdoor domain gap hypothesis and more carful analysis of underlying factors is needed.

**Weaknesses:**

-I am slightly concerned regarding how relevant this work is to NeurIPS Datasets and Benchmarks Track. Obviously, this is not a Dataset paper. Insights on important dataset attributes for 3D mesh recovery are only given through UMAP visualisations. Simple experimentation on optimal dataset mixtures and weighting schemes is also performed. Even though this work does audits on existing datasets (that is inside the score of NeurIPS Datasets and Benchmarks Track), the main focus is rather on identifying overlooked aspects of common pipelines for 3D mesh recovery (pretraining, augmentation, robustness to noise, appropriate losses).

-Section 3.3 provides an interesting analysis of the robustness of common architectures in the injection of controlled noise. The main motivation for this, is that many algorithms use pseudo-annotations during training. Such an analysis might not be representative of corruption in real-world pseudo-annotations. For example, an inaccurate body part will rarely be on a random location (as shown in Fig of the appendix) and inaccurate pseudo-labels are more likely on difficult samples (highly articulated poses or large occlusion).

-Experimental setting in section 3.3 might be a bit unclear. What is the x-axis on Fig2.b? Does the scale and location caption refer to the axis? Also, based on Fig2 how slight noise can help training?

-What do authors mean in line250, stating that transferring knowledge is not necessary to further finetune the backbone with ETF-COCO and how this is shown in Table 7?

-It’s interesting how weighting schemes can lead to increased performance. On the other hand, in the presented form this is a heuristic approach (manually selecting weights for multiple datasets) that increases method complexity.

**Additional Feedback:**

This work provides an interesting insight into overlook aspects of 3D human pose and shape estimation. I am slightly concerned about the relevance to NeurIPS Datasets and Benchmarks Track.

**Documentation:**

Since there isn't any resource introduced by this work, documentation is not required. Also Authors make available code for reproducing their experiments.

**Ethics:**

There are not ethical concerns regarding this work.

**Relation To Prior Work:**

Related work introducing the various components explored in this work is well presented.

**Summary And Contributions:**

This paper studies various components of 3D pose and shape estimation, that are usually overlooked or heuristically addressed by relevant work. A large-scale experimental study is performed that raises interesting insights on the impact of dataset mixing, importance of pretraining, effectiveness of various backbones, need for augmentation strategies, robustness to noise and others. Based on their finding, authors manage to train a strong baseline on 3DPW with a rather simple model. Multiple good practises for 3D pose and shape estimation are identified.

---

> ### Author Response · Authors · 2022-08-18
> **Response (1/3)**
>
> We sincerely thank the reviewer for your insightful comments and recognition to this work, especially for acknowledging how this work can serve as a valuable set of best practices for developing mesh recovery models. Below we provide point-to-point responses to address all the raised questions:
>
> > Q1: I am slightly concerned regarding how relevant this work is to NeurIPS Datasets and Benchmarks Track. Obviously, this is not a Dataset paper. Insights on important dataset attributes for 3D mesh recovery are only given through UMAP visualisations. Simple experimentation on optimal dataset mixtures and weighting schemes is also performed. Even though this work does audits on existing datasets (that is inside the score of NeurIPS Datasets and Benchmarks Track), the main focus is rather on identifying overlooked aspects of common pipelines for 3D mesh recovery (pretraining, augmentation, robustness to noise, appropriate losses)
>
> A1: We understand your concern. We believe that our paper can fit for the D&B track for the following reasons.
>
> First, we refer to the blog post referenced in the FAQs of the D&B track [https://neuripsconf.medium.com/announcing-the-neurips-2021-datasets-and-benchmarks-track-644e27c1e66c] and identified several scopes that are relevant to us:
> - **“audits of existing datasets, or systematic analysis of existing systems on novel datasets that yield important new insight are also in scope”** – We provide audits of datasets used in prior 3D pose and shape estimation studies and include new ones that have not been used (Table 2). We obtain new insights for what makes a dataset effective for training, which are helpful for future dataset selection, creation or enhancement of existing datasets.
> - **“As part of this track, we aim to gather advice on best practices in constructing, documenting, and using datasets”** – We provided advice on using and constructing datasets (Remark 1).
> - **“other benchmarking efforts to connect models to real world impacts”** – We identified optimal configurations that help model training.
>
> Second, there is precedence of similar works from NeurIPS’21 D&B track that do not provide new datasets or benchmarking metrics. For instance, similar to our paper, some papers [1, 2, 3] provide analysis on non-data aspects, i.e., architectures or how to build a more robust model; some papers evaluate the benchmarking practices [4, 5]. We believe those papers and ours all fit into this track.
>
> Third, model initialization and training strategies are relevant to dataset benchmarking, as the selection of those strategies can highly affect the selection of training datasets, as we discovered in this paper.
>
> > Q2: Section 3.3 provides an interesting analysis of the robustness of common architectures in the injection of controlled noise. The main motivation for this, is that many algorithms use pseudo-annotations during training. Such an analysis might not be representative of corruption in real-world pseudo-annotations. For example, an inaccurate body part will rarely be on a random location (as shown in Fig of the appendix) and inaccurate pseudo-labels are more likely on difficult samples (highly articulated poses or large occlusion.
>
> A2: Thank you for the insightful comment. We agree that some scenarios are more likely to occur i.e., inaccurate body parts might be more common in the end nodes and pseudo-labels are more likely on difficult samples.
>
> Firstly, we do have some considerations in simulating noise, our experiments are motivated by the fact that errors might occur in a portion of the sample or be localised in certain body parts. Therefore, we attempt to simulate noise in a controlled setting by perturbing it according to the (1) scale and (2) location.
>
> Secondly, it is a common practice to assume Gaussian noise in modeling real-life noises. We follow a few prior works that simulate noisy key points or poses. In [6], Gaussian noise is added to the (x, y) coordinates of the ground-truth signal. In [7], marker data is corrupted by  randomly removing and shifting markers to dynamically produce poses with errors similar to those found in real motion capture data.

---

> > ### Author Response · Authors · 2022-08-18
> > **Response (2/3)**
> >
> > > Q3: Experimental setting in section 3.3 might be a bit unclear. What is the x-axis on Fig2.b? Does the scale and location caption refer to the axis? Also, based on Fig2 how slight noise can help training?
> >
> > A3: We apologize for the confusion, and thank you for highlighting this. The scale and location description was meant to be the x-axis. We have revised the paper accordingly to make it more clear (Table 4).
> >
> > We have added extra clarification for scale and location below, and in L216 of Section 3.3 (marked in blue).
> > - “Scale”: Simple Gaussian Noise with different standard deviations following [6, 7] is added to all SMPL pose parameters (Fig 7a).
> > - “Location”: Cases for wrongly fitted body parts are simulated by replacing a percentage of pose parameters (body parts) with random noise (Fig 7b).
> >
> > The previous line “slightly noisy SMPL within realistic realms still helps training” might be misleading, so thank you for pointing this out. We do not mean that slight noise improves performance. In general, any amount of noise will increase errors and harm performance as seen from Fig 2. What we intend to express is that data containing slightly noisy SMPL annotations can still be useful for training. Errors from training with data containing a portion of noisy SMPL annotations (<50%) is still relatively low (Fig 2a - blue), implying that these samples can still be useful for training. We have corrected our wording in the revised paper in L224 (marked in blue).
> >
> > > Q4: What do authors mean in line250, stating that transferring knowledge is not necessary to further finetune the backbone with ETF-COCO and how this is shown in Table 7?
> >
> > A4: Sorry for the confusion. We have modified this part in Section 4 in the revised paper (marked in blue). What we intended to express is that transferring weights from a pose estimation model already brings large benefits. PARE is the first work to use backbone from a model trained on pose estimation tasks.  However, they only mentioned in their training details that the weights from a pose estimation task were used to later train on EFT-COCO for mesh recovery task. After which, the model trained on EFT-COCO was trained on the full mix of dataset. We found that this initialisation already leads to very good performance gains without having to take the extra step to fine tune on EFT-COCO.

---

> > > ### Author Response · Authors · 2022-08-18
> > > **Response (3/3)**
> > >
> > > > Q5: It’s interesting how weighting schemes can lead to increased performance. On the other hand, in the presented form this is a heuristic approach (manually selecting weights for multiple datasets) that increases method complexity.
> > >
> > > A5: Yes, we acknowledge that manual selection of datasets and partitions would increase method complexity. Training with multiple datasets is a problem rather unique to 3D pose and shape estimation, and most of the dataset selection and weighting is rather manual and ad-hoc currently. Therefore, automatic selection of datasets and weights would be important, and we will seriously consider this as future work. Below we discuss some hints about how to efficiently select the datasets, and how automatic selection could be done. These have been added to Section 3.2, Section 7 and Appendix A in the revised paper (marked in blue).
> > >
> > > 1. Although manual selection of datasets will increase complexity, we are not directionless. Our in-depth investigation provides a set of effective datasets that can be adopted directly in future research.
> > >
> > > 2. More importantly, our finding can be used to guide the development and evaluate the quality of new datasets. From our analysis, a good overlap in train-test distributions of features such as camera, pose, shape, backbone features would help to achieve good performance. We could then select the top N datasets that would cover a wide distribution. In addition, we identified several attributes of a dataset that makes it effective for training, such as SMPL annotations, and few datasets currently afford it. These findings could help us make informed choices on how to select datasets.
> > > 3. It is also interesting to have an automatic approach to dataset selection. In our paper, dataset-level weighting is more effective than sample-level weighting. If we consider dataset partition as a hyperparameter to tune, we can borrow methods from automatic hyperparameter tuning with methods such as reinforcement learning or bayesian optimization. This will be our future work.
> > >
> > > References:
> > >
> > > [1] Chen et al. “Benchmarks for Corruption Invariant Person Re-identification” NeurIPS 2021 Datasets and Benchmarks Track.
> > >
> > > [2] Li et al. "MQBench: Towards Reproducible and Deployable Model Quantization Benchmark." NeurIPS 2021 Datasets and Benchmarks Track.
> > >
> > > [3] Yi et al. “Benchmarking the Robustness of Spatial-Temporal Models Against Corruptions.” NeurIPS 2021 Datasets and Benchmarks Track.
> > >
> > > [4] Curth et al. “Really Doing Great at Estimating CATE? A Critical Look at ML Benchmarking Practices in Treatment Effect Estimation”. NeurIPS 2021 Datasets and Benchmarks Track.
> > >
> > > [5] Bao et al., “It’s COMPASlicated: The Messy Relationship between RAI Datasets and Algorithmic Fairness Benchmarks”.  NeurIPS 2021 Datasets and Benchmarks Track.
> > >
> > > [6] Gauss et al. Smoothing Skeleton Avatar Visualizations Using Signal Processing Technology. SN Computer Science 2021.
> > >
> > > [7] - Holden. Robust Solving of Optical Motion Capture Data by Denoising. ACM Transactions on Graphics 2018.
> > >
> > > Please don’t hesitate to let us know if there are any additional clarifications or experiments that we can offer!

---

> ### Author Response · Authors · 2022-08-25
> **Follow-up**
>
> Dear reviewer,
>
> We would like to follow up to check if your concerns have been addressed. In the previous response, we have made the following updates/clarification:
> - Regarding the fit of our paper for the D&B track (Q1), we have identified relevant scopes outlined in the D&B track, added precedence of similar works from NeurIPS’21 D&B track and added support for overlooked aspects such as augmentation and appropriate losses are also linked to datasets.
> - Regarding your comment on noise being biased to difficult samples or body parts, we acknowledged that this is often the case in real-world scenarios. In our revised paper, we added references to prior noise simulation experiments to justify our choice of using Gaussian noise (L218-219 in Section 3.3).
> - We have changed our wording and added clarification in our revised paper following your suggestions in Q3 (Figure 4) and Q4 (L228).
> - Regarding your concern on manual weighting increases complexity, we demonstrate that we are not directionless and have added possible approaches for future works.
>
> We are happy to answer further questions.

---

### Official Review · Reviewer_BCuk · 2022-07-27
**A comprehensive benchmark for 3D human pose and shape estimation**

**Rating:** 2
**Confidence:** 5
**Clarity:** Yes. The paper is well written and ea…

**Strengths:**

This paper provides extensive experimental results for the task of 3D human pose and shape estimation, which are well organized and easy to follow. This can save time for other researchers who are curious about this research field from a practical perspective.

**Weaknesses:**

1. Although the time and effort spent by the authors on all the experiments is impressive, the contribution of this paper is questionable. All the remarks are obvious to the researchers in this field even without any of the experiments. It is good to know the numbers and rankings given by the experiments, but that is all. There is nothing more meaningful that can be learnt from this paper.
2. It is helpful to put all the datasets together and show some statistics, but all the datasets are kept as is, and nothing new is introduced. The experiment of different training set is strongly biased by the HMR method, as well as the 3DPW and H36M test set. The conclusions drawn from this experiment is ad-hoc.
3. The backbone architecture and pre-training is another problem, which has nothing specific to do with the task of 3D human pose and shape estimation.
4. Data augmentation is a common practice when there is limited training data. There is nothing new about data augmentation in this paper.
5. Training loss selection depends on the method and the data. The difference between MSE and L1 loss is already known. No additional insight is provided in this paper.


**Additional Feedback:**

I admire the time and effort that the authors spent on all the experiments. The experiments may be helpful to someone, but not to the serious researchers in this field.

**Correctness:**

Yes. The claims made in the submission are correct. The evaluation methods and experiment design are appropriate and performed correctly.

**Documentation:**

Yes. There is sufficient detail to support reproducibility.

**Ethics:**

No. There is no ethical concern.

**Relation To Prior Work:**

Yes. The difference of this work from previous works is clearly discussed.

**Summary And Contributions:**

This paper experimentally studies the task of 3D human pose and shape estimation from three aspect: dataset, backbone, and training strategy. Specifically, this paper provides a detailed analysis of 31 datasets on their data characteristics, as well as how they affect the performance when used for training. The authors experiment with 10 different neural network architectures as backbones and compare their performance. The dataset used for pre-training the backbone is also experimentally shown to be important. Moreover, training strategies including data augmentation and training loss are discussed. Although the time and effort spent on the experiments is significant, no meaningful insight is given by the authors.

---

> ### Author Response · Authors · 2022-08-18
> **Response (1/3)**
>
> We sincerely thank the reviewer for your insightful and constructive feedback. The technical contributions and novelty of this work are highlighted in the General Response. Please kindly refer to it for details. Below we would like to provide point-to-point responses to address all the raised questions
>
> > Q1: Although the time and effort spent by the authors on all the experiments is impressive, the contribution of this paper is questionable. All the remarks are obvious to the researchers in this field even without any of the experiments. It is good to know the numbers and rankings given by the experiments, but that is all. There is nothing more meaningful that can be learnt from this paper.
>
> A1: We would like to re-emphasise the technical contributions and novelty of this work in the General Response. **We provided a systematic study of factors beyond algorithms that influence the performance of 3D pose and shape estimation. We believe that our findings and baselines are useful to the researchers in 3D mesh recovery, as prior works used much less optimal baselines and performed unfair comparisons.** We add a new section to summarize the key findings in Appendix A, which will be helpful to the researchers to build strong models. We are also glad that our contributions are acknowledged by the other five reviewers.
>
> > Q2: It is helpful to put all the datasets together and show some statistics, but all the datasets are kept as is, and nothing new is introduced. The experiment of different training set is strongly biased by the HMR method, as well as the 3DPW and H36M test set. The conclusions drawn from this experiment is ad-hoc.
>
> A2: Thank you for the comments. 3d pose and human shape estimation is a unique problem where multiple datasets are trained together in a mixture without dissecting the impact of individual datasets. **The selection of dataset is often ad-hoc and the contribution of individual datasets in the dataset mix has not been investigated.** From our benchmarks on individual datasets, we gained several insights into what makes a dataset effective or ineffective for training (Remark 1 and Appendix A) . For instance, fitting SMPL annotations could greatly improve the effectiveness of widely available 2D keypoint datasets for 3D mesh recovery tasks.  High diversities in the distribution of certain attributes (i.e. human pose, shape, camera characteristics and backbone features), leading to considerable overlap between training and test set distributions are critical. We believe that these insights could be valuable for other researchers to (1) select datasets for training (2) creating/ collecting new datasets (3) enhancing existing datasets i.e. improve the less effective datasets by adding SMPL annotations.
>
> To make our evaluations more comprehensive, we follow your advice to add more experiments with different algorithms (SPIN [1], GraphCMR [2], PARE [3], Graphormer [4]) and different test sets. Our findings are general to different algorithms and test sets. Please check Section 6 (Tables 10 and 11)  for more detailed descriptions and analysis. We have also included the results below:
>
> | Algorithms | Datasets   | Backbone | Initialisation | Normal | L1 | L1+COCO | L1+COCO+Aug |
> |:------:|:-------:|:------:|:-------:|:------:|:------:|:-------:|:------:|
> | HMR | H36M, MI, COCO, LSP, LSPET, MPII | ResNet-50 | ImageNet | 64.55 | 58.20 | 51.8 | 51.66 |
> | SPIN [1] | H36M, MI, COCO, LSP, LSPET, MPII | ResNet-50 | HMR(ImageNet) | 59.00 | 57.08 | 51.54 | 50.69 |
> | GraphCMR [2] | H36M, COCO, LSP, LSPET, MPII, UP3D | ResNet-50 | ImageNet | 70.51 | 67.2 | 61.74 | 60.26 |
> | PARE [3] | H36M, MI, EFT-[COCO, LSPET, MPII] | HRNet-W32 | ImageNet | 61.99 | 61.13 | 59.98 | 58.32 |
> | Graphormer [4] | H36M, MuCo, COCO, UP3D, MPII | HRNet-W48 | ImageNet | 63.18 | 63.47 | 59.66 | 58.82 |

---

> > ### Author Response · Authors · 2022-08-18
> > **Response (2/3)**
> >
> > > Q3: The backbone architecture and pre-training is another problem, which has nothing specific to do with the task of 3D human pose and shape estimation.
> >
> > A3: We want to clarify that **the selection of backbone architecture and pre-training is highly related to the performance of 3D human pose and shape estimation, which is rarely investigated in prior works.**
> >
> > 1. For backbone architecture, we find previous works compare performance with other algorithms even when different backbone architectures are used, which could lead to unfair evaluation. Our paper systematically benchmarks the impact of different backbones, which can promote fair evaluations. In addition, prior works have yet to explore the impact of contemporary discoveries in the community i.e. transformers. HRNet is often preferred over the ResNet series in the field of human pose estimation, and we validate the same for human pose and shape estimation (Table 6). As more recent works switch to HRNet variants [3, 4], transformers have yet to be applied to 3D pose and shape estimation tasks.  Our work finds that transformers tend to perform better than CNNs at comparable FLOPs, opening the avenue for the field to switch towards better feature extractors. Our investigation can help the industry to make an informed choice, and select the most suitable backbone in actual application with optimal accuracy-computation cost trade off.
> >
> > 2. For pre-training, the effectiveness of transferring knowledge from pose estimation to mesh recovery task has not been investigated despite the effectiveness in model performance (Table 7). To the best of our knowledge, PARE is the first work to use backbone from a model trained on pose estimation tasks. However, this was only mentioned as a training detail. Our work provides the insight that knowledge from pose estimation is highly complementary for 3D mesh recovery tasks. This indicates that we can adopt readily available pre-trained pose estimation as initialisation to achieve faster convergence or build stronger mesh recovery models.
> >
> > > Q4: Data augmentation is a common practice when there is limited training data. There is nothing new about data augmentation in this paper.
> >
> > A4: We want to clarify that although data augmentation is a common practice in conventional CV tasks, **it is less explored in the human mesh recovery task, especially in the multi-dataset setting. We present the first systematic study towards the effectiveness of different augmentations on individual datasets when they are trained together for 3d pose and human shape estimation.** We find that certain augmentations are more effective than others (Table 8) and offer possible explanations. Directly applying augmentation might bring more harm than good, as EFT [5] reported marginal improvement and PARE [3] reported a worsening in performance when applying crop augmentation. We believe it might have to do with distribution shifts in some of the training datasets. We add more discussions regarding this as future work in Section 7 of the revised paper. In addition, we propose selective augmentation, taking care to augment datasets that lack diversity (augmentation would therefore be beneficial) while avoiding distribution shift in datasets with already diverse distributions. We also perform in-depth analysis according to the train-test set distributions and make the recommendation for selective augmentation in a multi-dataset setting. This is not explored by previous works on human mesh recovery.

---

> > > ### Author Response · Authors · 2022-08-18
> > > **Response (3/3)**
> > >
> > > > Q5: Training loss selection depends on the method and the data. The difference between MSE and L1 loss is already known. No additional insight is provided in this paper.
> > >
> > > A5: 3D pose and shape estimation is a complex problem where often different losses are computed for different supervision of vertex, keypoints, smpl parameters. In our review of different algorithms and the losses they used (Table 12 in appendix), **the majority of mesh recovery models use MSE loss for regression of keypoints [30, 56, 35, 36, 84, 29, 37, 12, 33], which give less optimal performance. This indicates that the difference between MSE and L1 loss is not widely adopted in this community.** We are the first to systematically demonstrate the effectiveness of L1 on different dataset mixes (Table 9) and show that it can curb noisy SMPL annotations (Figure 4), which makes it useful for the task for 3D pose and shape estimation when SMPL labels are important.
> > >
> > > References:
> > >
> > > [1] Kolotouros et al. “Learning to Reconstruct 3D Human Pose and Shape via Model-fitting in the Loop.” ICCV 2019.
> > >
> > > [2] Kolotouros et al. “Convolutional Mesh Regression for Single-Image Human Shape Reconstruction”. CVPR 2019.
> > >
> > > [3] Kocabas et al. “PARE: Part Attention Regressor for 3D Human Body Estimation.” ICCV 2021.
> > >
> > > [4] Lin et al. “Mesh Graphormer”. ICCV 2021.
> > >
> > > [5] Joo et al. “Exemplar Fine-Tuning for 3D Human Pose Fitting Towards In-the-Wild 3D Human Pose Estimation.” 3DV 2021.
> > >
> > > Please don’t hesitate to let us know if there are any additional clarifications or experiments that we can offer!

---

> > > > ### Comment · Reviewer_BCuk · 2022-08-20
> > > > **Additional comments**
> > > >
> > > > I would like to thank the authors for their response and additional experiments. However, there are already plenty of experiments, and I do not worry about the scale of the experiment in this paper. My concern is about the motivation. I would like to see in-depth thinking about the problem instead of more experiments.
> > > > 1. The authors fail to precisely define “3D human pose and shape estimation” and “human mesh recovery”. Does it specifically refer to paper [30], which is influential but outdated? Or does it refer to related works that may have different setups? If so, what are the specific setup and assumptions that are used by this paper?
> > > > 2. This paper only uses PA-MPJPE as the evaluation metric, which just measures the error for body joints, and has nothing to do with the body shape or mesh surface. It does not make sense to emphasize “shape estimation” and “mesh recovery” without defining any error measurement for them.
> > > > 3. In terms of the motivation, “prior works used much less optimal baselines and performed unfair comparisons” could not convince me. This is an activate research area, and people are still exploring different directions without achieving a consensus on what the ultimate goal is. It does not make sense to have “optimal baselines” and “fair comparisons” for a problem that is not well-established. If you are trying to build a dedicated baseline for the setup of paper [30], it does not make sense either, as it is the first attempt on this problem and it is outdated.
> > > > 4. In terms of the dataset experiments, “3d pose and human shape estimation is a unique problem” is not a valid justification. What is the uniqueness of the problem that invalidates the conclusion from the literature? Remark 1 is mainly about data diversity, which is exactly what we learnt from all the other dataset experiments.
> > > > 5. In terms the backbone experiments, changing backbone will almost certainly affect the performance of networks for different tasks. But it is not clear what is special here.
> > > > 6. For pre-training, the conclusion is that transferring knowledge from pose estimation is helpful. However, “pose estimation” is exactly what you are doing according to your evaluation metric PA-MPJPE.
> > > > 7. For data augmentation, "it is less explored in the human mesh recovery task” does not justify your motivation. What is special about this task that persuade you to do the experiments? As for Remark 5, the claim “data augmentation depends on training set” may not be helpful for future research.
> > > > 8. In terms of training using MSE vs L1 loss to achieve “optimal performance”, it is too trivial in this fast-evolving research area. The conclusion is not surprising, either.

---

> > > > > ### Author Response · Authors · 2022-08-25
> > > > > **Response (1/3)**
> > > > >
> > > > > We sincerely thank the reviewer for your insightful and constructive feedback. Below we would like to provide point-to-point responses to address all the raised questions
> > > > >
> > > > > > Q: I would like to thank the authors for their response and additional experiments. However, there are already plenty of experiments, and I do not worry about the scale of the experiment in this paper. My concern is about the motivation. I would like to see in-depth thinking about the problem instead of more experiments.
> > > > >
> > > > > A: Thank you for the suggestion. We would like to re-emphasize our motivations:
> > > > >
> > > > > a) Developing an effective vision model involves several critical aspects, including datasets, backbones, model initialisation and training strategies. Each aspect would have a large impact on the final performance. That's why our community is performing not only model-centric investigation, but also data-centric investigation [1]. It is also the reason that NeurIPS sets up the datasets and benchmarks track.
> > > > >
> > > > > b) However, existing works in 3D pose and shape estimation mainly focus on algorithm design or model-centric investigation. It has two potential drawbacks: i) the other important aspects are not fully understood, leading to suboptimal recipes, ii) different methods adopt different protocols (dataset mixes, backbones, training strategies), leading to unfair comparison. Therefore, it is necessary to have a comprehensive investigation on factors beyond algorithms that influence performance of 3D pose and shape estimation models.
> > > > >
> > > > > c) The computer vision community already benefits from this type of study. For example, i) the famous “devils are in the details” series [2], ii) benchmarking training strategies in image classification [3] and object detection [4]
> > > > >
> > > > > d) We hope our work could also unveil critical aspects and promote fair evaluations in the 3D human pose and shape estimation community.
> > > > >
> > > > > > Q1: The authors fail to precisely define “3D human pose and shape estimation” and “human mesh recovery”. Does it specifically refer to paper [30], which is influential but outdated? Or does it refer to related works that may have different setups? If so, what are the specific setup and assumptions that are used by this paper?
> > > > >
> > > > > A1: 3D pose and shape estimation are well known terminologies in this community. They do not specifically refer to paper [30]. They refer to a general task, which predicts the parameters of a statistical human body model or vertices from single RGB images or monocular RGB videos [35, 12, 37, 32, 33, 62, 59, 48, 34, 64, 60, 58, 29, 5, 12, 43, 73, 44]. We follow the same setup and assumptions to address the same task in the above works, which are very common practice.
> > > > >
> > > > > > Q2: This paper only uses PA-MPJPE as the evaluation metric, which just measures the error for body joints, and has nothing to do with the body shape or mesh surface. It does not make sense to emphasize “shape estimation” and “mesh recovery” without defining any error measurement for them.
> > > > >
> > > > >
> > > > > A2:  Thanks for your comments. PA-MPJPE is a standard metric widely used in prior 3d pose and shape estimation works [57, 36, 18, 35, 24, 87, 8, 15, 53, 84, 54, 40, 43, 85, 81, 29, 37, 12, 71, 44, 82, 83, 34, 33]. The choice of PA-MPJPE in our field is reasonable, because our goal is to infer accurate $\beta$ and $\theta$ parameters, which are later taken as input for parametric human models to get joint locations. This metric has already implied the evaluation of human shape and mesh [44, 73, 36, 43]. To make our evaluations more comprehensive, we added more metrics such as PVE, PA-PVE and MPJPE to Tables 1, 2, 4, 6 of the revised paper following [5, 6, 7, 8] and clarification in L88-93. We confirm that our findings are consistent when using other evaluation metrics, and  models with the lowest PA-MPJPE errors also have the lowest errors on other metrics (MPJPE/ PVE and PA-PVE).
> > > > >
> > > > > > Q3a: In terms of the motivation, “prior works used much less optimal baselines and performed unfair comparisons” could not convince me. This is an activate research area, and people are still exploring different directions without achieving a consensus on what the ultimate goal is. It does not make sense to have “optimal baselines” and “fair comparisons” for a problem that is not well-established.
> > > > >
> > > > > A3a: We respectfully disagree with you on this point. We think **it is rather important to have fair comparisons and evaluations especially when the problem is not fully established, when the community does not have an in-depth understanding of the impact of various factors (algorithm, backbone and initialisation, training strategies etc.) which are rather influential in model performance**. Without fair comparisons, it is hard for researchers to get the correct directions to improve the model and make the problem established.

---

> > > > > > ### Author Response · Authors · 2022-08-25
> > > > > > **Response (2/3)**
> > > > > >
> > > > > >
> > > > > > > Q3b:  If you are trying to build a dedicated baseline for the setup of paper [30], it does not make sense either, as it is the first attempt on this problem and it is outdated.
> > > > > >
> > > > > > First, as we mentioned previously, our work is not just for [30], but for many mesh recovery studies [35, 12, 37, 32, 33, 62, 59, 58, 29, 5]. Second, we respectfully disagree that HMR [30] is an “outdated” model. Many recent works [42, 34, 33, 12, 83, 29] still build upon the HMR framework. Third, our findings on HMR are generalisable to other works. We follow your advice to add more experiments with different algorithms (SPIN [35], GraphCMR [36], PARE [33], Graphormer [44]) and different test sets (Tables 10 and 11). Our findings are consistent with different configurations.
> > > > > >
> > > > > > HMR has been used in the performance comparison of many works [57, 36, 18, 35, 24, 87, 8, 15, 53, 84, 54, 40, 43, 85, 81, 29, 37, 12, 71, 44, 82, 83, 34, 33] as a suboptimal model. However, we demonstrated that with more recent dataset mixes (Table 4) and optimized configurations (Table 1), HMR can even perform on par to many SOTA models. Our study provided updated benchmarks on HMR which can serve as strong baselines for future performance comparisons.
> > > > > >
> > > > > > > Q4: In terms of the dataset experiments, “3d pose and human shape estimation is a unique problem” is not a valid justification. What is the uniqueness of the problem that invalidates the conclusion from the literature? Remark 1 is mainly about data diversity, which is exactly what we learnt from all the other dataset experiments.
> > > > > >
> > > > > > A4: 3d pose and shape estimation has several unique differences: 1) 3D pose and shape estimation models are generally trained over multiple datasets 2) Datasets used in 3D pose and shape estimation tasks typically lack paired data due to the prohibitive cost of data collection and annotation of 3D humans (especially in the wild). Therefore, multiple datasets of different modalities are used (2D keypoints, 3D keypoints, SMPL annotations, segmentation maps). Meanwhile, The biggest uniqueness is that 3D pose and shape estimation models are generally trained over multiple datasets, while many computer vision tasks often have a consistent training and benchmarking dataset, i .e., image classification (ImageNet, CIFAR), semantic segmentation (ADE20K, Citiscapes, PASCAL VOC), object detection (COCO, PASCAL VOC), 2D pose estimation (MPII, COCO, LSP). Because of such uniqueness, the choice of datasets can have a large influence on performance (Table 4), which is rarely explored in other CV tasks. Besides, there are also other important findings in Remark 1, e.g., the impact of SMPL and keypoint annotations, person-person occlusion for indoor datasets, which are unique and do not need to be considered in other CV tasks.
> > > > > >
> > > > > > We hope you can provide some references as to which “literature” and “all other dataset experiments” you refer to. We think that this would greatly help us to compare our work and better answer your concerns.
> > > > > >
> > > > > > > Q5: In terms the backbone experiments, changing backbone will almost certainly affect the performance of networks for different tasks. But it is not clear what is special here.
> > > > > >
> > > > > > A5: We think that it should be a practice to highlight backbone choice during performance comparisons to other algorithms. Our investigation can help the industry to make an informed choice, and select the most suitable backbone in actual application with optimal accuracy-computation cost trade off. This might inform practitioners, which is also in line with the scope of the D&B track to connect models to real world impacts.
> > > > > >
> > > > > > > Q6: For pre-training, the conclusion is that transferring knowledge from pose estimation is helpful. However, “pose estimation” is exactly what you are doing according to your evaluation metric PA-MPJPE.
> > > > > >
> > > > > > A6: 3D pose and shape estimation and 2D pose estimation are fundamentally two different tasks. The former estimates parameters to a 3D statistical human model, while the latter predicts the locations of 2D keypoints on an image.
> > > > > >
> > > > > > For pre-training, a bigger point that we would like to draw is that weight initialization has a large influence on performance (Table 7). Therefore, we think that it should be a practice to highlight weight initalization during performance comparisons to other algorithms.

---

> > > > > > > ### Author Response · Authors · 2022-08-25
> > > > > > > **Response (3/3)**
> > > > > > >
> > > > > > > > Q7: For data augmentation, "it is less explored in the human mesh recovery task” does not justify your motivation. What is special about this task that persuade you to do the experiments? As for Remark 5, the claim “data augmentation depends on training set” may not be helpful for future research.
> > > > > > >
> > > > > > > A7: As explained in A4, the special feature about this task is that 3D pose and shape estimation models are generally trained over multiple datasets. As a result, effects of different augmentation can vary significantly between datasets, and different datasets need different augmentation operations. This is rarely considered in other CV tasks, where the models are usually trained over one consistent dataset.
> > > > > > >
> > > > > > > The support for our claim “data augmentation depends on the training set” is demonstrated in Table 8.  The augmentation analysis has several benefits:
> > > > > > >
> > > > > > > - It gives some insights as to what indoor datasets are lacking. Specifically, person-person occlusion is most effective to help to enhance the dataset.
> > > > > > > - Perhaps we should not apply augmentation wholescale, and consider selective augmentation to avoid distribution shift.
> > > > > > >
> > > > > > > > Q8: In terms of training using MSE vs L1 loss to achieve “optimal performance”, it is too trivial in this fast-evolving research area. The conclusion is not surprising, either.
> > > > > > >
> > > > > > > A8: As mentioned in the previous response, this finding is novel and useful to human pose and shape estimation because (1) few works have adopted it despite its effectiveness, majority of the work uses MSE loss of keypoints and/or SMPL supervision [57, 36, 35, 8, 53, 29, 37, 12, 33] (2) we conducted noise experiments to justify that it helps to curb noisy SMPL labels and keypoints, which are specific to the field of pose and shape estimation (Figure 4). This makes it useful for the task for 3D pose and shape estimation when SMPL labels are important.
> > > > > > >
> > > > > > > References:
> > > > > > >
> > > > > > > [1] Mazumder et al. “DataPerf: Benchmarks for Data-Centric AI Development”. arXiv 2022.
> > > > > > >
> > > > > > > [2] Chatfield et al. “Return of the Devil in the Details: Delving Deep into Convolutional Nets”. BMVC 2014
> > > > > > >
> > > > > > > [3] He at al “Bag of Tricks for Image Classification with Convolutional Neural Networks”. arXiv 2018.
> > > > > > >
> > > > > > > [4] Zhang et al. “Bag of Freebies for Training Object Detection Neural Networks”. arXiv 2019.
> > > > > > >
> > > > > > > [5] Li et al. “Everybody Is Unique: Towards Unbiased Human Mesh Recovery”. BMVC 2021.
> > > > > > >
> > > > > > > [6] Zhang et al “PyMAF: 3D Human Pose and Shape Regression with Pyramidal Mesh Alignment Feedback Loop” ICCV 2021.
> > > > > > >
> > > > > > > [7] Lin et al. “End-to-End Human Pose and Mesh Reconstruction with Transformers”.  CVPR, 2021
> > > > > > >
> > > > > > > [8] Lin et al. “Mesh Graphormer”. ICCV 2021.
> > > > > > >
> > > > > > > Once again,  we hope that our responses can help to clarify the merits of this paper. Please don’t hesitate to let us know if there are any additional clarifications or experiments that we can offer.

---

> > > > > > > > ### Comment · Reviewer_BCuk · 2022-08-29
> > > > > > > > **Comments**
> > > > > > > >
> > > > > > > > Thanks for the further clarification. I agree that the format of this study perfectly matches the expectation of NeurIPS datasets and benchmarks track. However, it should not be an excuse for the lack of proper problem definition, motivation, and in-depth discussion, especially when the experimental conclusions are superficial.
> > > > > > > > 1. The authors’ motivation still seems vague to me, and I do not see the authors’ thinking on the “3D pose and shape estimation” task itself. The justification is task-agnostic, and it seems that the authors can do exactly the same for any task.
> > > > > > > > 2. Thanks for the clarification. I do not agree that “3D pose and shape estimation” specifically means estimating parameters for parametric models like SMPL. That is why I was asking about paper [30], and unfortunately this is exactly what the authors are doing based on the clarification. Parametric body models like SMPL are useful in that they can reasonably constrain the ill-posed 2D-to-3D problem. However, their shape parameters are an under-parameterization of the actual body shapes, and most variations in actual body shapes cannot be properly represented. As a consequence, accurately estimating SMPL shape parameters $\beta$ can never be our ultimate goal. Estimating pose parameters $\theta$ makes sense, though, as they are an over-parameterization.
> > > > > > > > 3. Since the authors’ goal is to infer SMPL parameters, I agree that PA-MPJPE measures the effect of both parameters. However, if the authors really want to have an in-depth understanding of the problem, it makes more sense to study each parameter separately.
> > > > > > > > 4. As I commented above, estimating accurate $\beta$ should not be our final goal, and there is not even ground-truth $\beta$ for real-world data, as the SMPL model is not expressive enough. Existing methods have shown the feasibility of this task, and a rough estimation of SMPL parametric model makes it possible for further down-stream tasks. However, it does not make sense to have a super “accurate” SMPL estimation, since there is only pseudo-ground-truth except for synthetic data. I agree that people do not have a good understanding of the problem yet, but the conclusions in this paper could not help people better understand this problem, either. What is worse, this paper may encourage researchers to overfit to this ill-posed problem.
> > > > > > > > 5. When I was asking about paper [30], I meant the parametric body model estimation. According to the authors’ clarification, this is exactly what they are doing. HMR has its own value by showcasing the feasibility of this approach. However, this approach is bounded by the representation of SMPL. I do not think it is a good practice to encourage researchers to overfit to this ill-posed setting.
> > > > > > > > 6. Training over multiple datasets is a practice when there is not sufficient training data. It is task-agnostic, so people can do it for any task when they think they need more training data. I still could not see the authors’ thinking on the “3D pose and shape estimation“ problem itself. Well-known dataset papers like ImageNet and COCO have extensively discussed the importance of data diversity. For the “3D pose and shape estimation” task, specifically, the variation of the data should come from whatever characteristics that affect the observation of body pose and shape. The experiments in the paper just confirmed this intuitive conclusion.
> > > > > > > > 7. There is no denying that the choice of backbone affects the performance. However, I could not see in-depth discussion on the backbone selection in the context of the ”3D pose and shape estimation” task. It is not clear if the conclusion can be generalized to other models for the same task.
> > > > > > > > 8. I agree that 3D tasks and 2D tasks are fundamentally different, but they also have some connections for “pose estimation”. It would be nice to see the authors’ in-depth discussion on this in the paper.
> > > > > > > > 9. See 6 for my comments on the authors’ justification. For the highlighted benefits, 1) the occlusion trick has been studied in paper [29]; 2) “selective augmentation” is not precisely defined, hence impractical.
> > > > > > > > 10. This finding is not novel. It is a common sense that MSE loss makes the network converge faster, but it is more sensitive to noise comparing to L1 loss. This knowledges is task-independent. People just use arbitrary loss to showcase the feasibility of their methods, and it does not make sense to pursue "optimal performance" for this ill-posed problem (see my comment 2 and 4).

---

> > > > > > > > > ### Author Response · Authors · 2022-08-29
> > > > > > > > > **Response**
> > > > > > > > >
> > > > > > > > > We thank the reviewer for the detailed comments. We believe your concerns mainly stems from the misunderstanding of the 3D pose and shape estimation task.
> > > > > > > > >
> > > > > > > > > First, we clarify that we do not try to rethink the problem definition for 3D pose and shape estimation. The problem of 3D pose and shape estimation has been defined in Lines 20-24. Our motivations are also well discussed in Section 1. We want to emphasize again that **“3D human pose and shape estimation” is a well-defined problem in this community.** We follow the common settings and assumptions to benchmark datasets, backbones and training strategies.
> > > > > > > > >
> > > > > > > > > Second, we think that you have a misunderstanding of the term “3D pose and shape estimation”. According to the entire line of work in this field [56, 18, 35, 24, 86, 8, 15, 52, 83, 84, 80, 29, 47, 12, 70, 33, 36, 43], the term 3D pose and shape estimation has referred to estimating the “pose” and “shape” parameters of a statistical human body model. Moreover,  in [2] the SMPL body model is introduced to be a “pose and shape model”.
> > > > > > > > >
> > > > > > > > > Third, we would like to re-emphasise that the purpose of our study is not to debate about the feasibility of SMPL body models but rather, to promote fair evaluations and provide strong baselines for a large community that works on 3D pose and shape estimation using parametric body models. In addition, we would like to highlight that the use of SMPL/ parametric body model is a widely recognized research direction which draws increasing attention from the community [1]. More importantly, newer parametric models [3, 4], datasets [5] and methods [6] are looking to address the limitations of parametric models (i.e. no ground-truth, poor shape representation) that you mentioned.
> > > > > > > > >
> > > > > > > > > Below we provide point-to-point responses:
> > > > > > > > >
> > > > > > > > > > Q1: Training over multiple datasets is a practice when there is not sufficient training data. It is task-agnostic, so people can do it for any task when they think they need more training data… For the “3D pose and shape estimation” task, specifically, the variation of the data should come from whatever characteristics that affect the observation of body pose and shape.
> > > > > > > > >
> > > > > > > > > A1: We highlight that selection of training datasets is not trivial.
> > > > > > > > > 1) More data does not necessarily mean better performance (Table 4), simply adding more data leads to worsened performance.
> > > > > > > > > 2) Some datasets are more effective than others (Table 2), this is especially important in case of limited computational budget.
> > > > > > > > > 3) When training with multiple datasets, we also need to consider their relative contribution (Table 5) and selective augmentation (Table 8) for maximum benefits.
> > > > > > > > >
> > > > > > > > > > Q2: There is no denying that the choice of backbone affects the performance. However, I could not see in-depth discussion on the backbone selection in the context of the ”3D pose and shape estimation” task. It is not clear if the conclusion can be generalized to other models for the same task.
> > > > > > > > >
> > > > > > > > > A2: We have added additional experiments for optimized configurations on the other algorithms in the revised paper (Table 16). In general, HRNet and Twins-SVT perform better than ResNet-50.
> > > > > > > > >
> > > > > > > > > > Q3: For the highlighted benefits, 1) the occlusion trick has been studied in paper [29]; 2) “selective augmentation” is not precisely defined, hence impractical.
> > > > > > > > >
> > > > > > > > > A3: Paper [29] specifically studies crop-augmentation. In Section 5.1, we have benchmarked additional 8 image-based augmentation on top of crop augmentation on two datasets of vastly different domains. In addition, PARE [33] adopted crop augmentation [29] and found that it did not improve model performance, which prompts us to investigate the effect of augmentation  in a multi-dataset setting.
> > > > > > > > >
> > > > > > > > > We refer to selective augmentation literally to augment specific datasets. This technology has already been practically used in image recognition as “learning to augment” or “AutoAug” [7].
> > > > > > > > >
> > > > > > > > > > Q4: This finding is not novel. It is a common sense that MSE loss makes the network converge faster, but it is more sensitive to noise comparing to L1 loss. This knowledges is task-independent. People just use arbitrary loss to showcase the feasibility of their methods.
> > > > > > > > >
> > > > > > > > > A4: Please refer to A8 in our previous response on L1 loss.
> > > > > > > > >
> > > > > > > > > References:
> > > > > > > > >
> > > > > > > > > [1] Tian et al. “Recovering 3D Human Mesh from Monocular Images: A Survey”. arXiv 2022.
> > > > > > > > >
> > > > > > > > > [2] Loper et al. “SMPL: A Skinned Multi-Person Linear Model Matthew”. ACM Transactions on Graphics.
> > > > > > > > >
> > > > > > > > > [3] Osman et al. “STAR: A Sparse Trained Articulated Human Body Regressor”. ECCV 2021.
> > > > > > > > >
> > > > > > > > > [4] Xu et al. “GHUM & GHUML: Generative 3D Human Shape and Articulated Pose Models”. CVPR 2020.
> > > > > > > > >
> > > > > > > > > [5] Cai et al. “HuMMan: Multi-Modal 4D Human Dataset for Versatile Sensing and Modeling”. ECCV 2022.
> > > > > > > > >
> > > > > > > > > [6] Choutas et al. “Accurate 3D Body Shape Regression using Metric and Semantic Attributes”. CVPR 2022.
> > > > > > > > >
> > > > > > > > > [7] Cubuk et al. “AutoAugment: Learning Augmentation Strategies from Data”. CVPR 2019

---

### Official Review · Reviewer_gtaS · 2022-07-27
**Review for 3D Human Pose and Shape Estimation Benchmarking Study**

**Rating:** 5
**Confidence:** 4
**Clarity:** The paper is well written, without ob…

**Strengths:**

1. The problem is properly motivated and studies a well-known field of human mesh recovery
2. Experiments are thorough and reasonable
3. The recommendations provided by the authors via the benchmarking study will be helpful for future researchers to build strong models
4. Strong baselines will provide a better baseline for fair future comparisons

**Weaknesses:**

1. The authors use HMR as the baseline. Many improved models have come out since then, which could have been better choices.
2. While the paper claims to "provide strong baselines for fair comparisons of algorithms," baselines are only done for one model (HMR).
3. The paper does not present a dataset or a benchmark, but rather proposes stronger baselines on existing benchmarks, and recommendations for training strategies - it's not clear that this is in scope for the D&B track.


**Additional Feedback:**

Since your goal is to provide a "strong baselines for fair comparisons of algorithms" I think it would be reasonable to provide more than 1 baseline (HMR), which is currently the case. The paper gradually builds towards finding the optimal dataset, model parameters, and training method, so as a final conclusion, I would have liked to seen a set of baselines on many models besides HMR.

**Correctness:**

The paper discusses multiple components of HMR training, each of which is supported with experiments. The experimental design is appropriate and performed correctly, with extensive experiments providing convincing support for the recommendations made by the authors.

**Documentation:**

Yes, the authors provide code for their benchmarking study.

**Ethics:**

No new data is introduced. No other ethical concerns.

**Relation To Prior Work:**

Yes, this paper is the first large-scale benchmarking study of human mesh recovery.

**Summary And Contributions:**

The authors tackle the problem of extensively evaluating human mesh recovery model and training parameters to find the optimal parameters for a strong baseline. They perform extensive ablation studies on many components of the model, including data type, data augmentation, model choice, model initialization, and training strategy. The authors achieve improved results on 3DPW.

---

> ### Author Response · Authors · 2022-08-18
> **Response (1/2)**
>
> We sincerely thank the reviewer for your insightful comments and recognition to this work, especially for extensive ablation studies on many components of the model, including data type, data augmentation, model choice, model initialization, and training strategy. Below we would like to provide point-to-point responses to address all the raised questions
>
> > Q1: The authors use HMR as the baseline. Many improved models have come out since then, which could have been better choices.
>
> A1: Thank you for the advice! We would like to clarify that our choice of HMR was motivated by the fact that the origin of many mesh recovery works [35, 12, 37, 32, 33, 61, 58, 47] can be traced back to HMR [29]. Many notable works are built upon HMR i.e. SPIN [35]  added an optimization loop, SPEC [34] estimating camera parameters, ProHMR [37] using probabilistic estimation to derive the pose. It has also been widely used as the baseline in many studies [57, 29, 5, 12].
>
> Following your suggestion, we also run our major findings on additional network models (SPIN [1], GraphCMR [2], PARE [3], Graphormer [4]) into the evaluation in our revision. Please check Section 6 (Table 10) for more detailed descriptions and analysis. In brief, **our evaluation approaches are general to different algorithms, and the conclusions we obtained from HMR are consistent with other algorithms as well.**
> We have added the results below:
> | Algorithms | Datasets   | Backbone | Initialisation | Normal | L1 | L1+COCO | L1+COCO+Aug |
> |:------:|:-------:|:------:|:-------:|:------:|:------:|:-------:|:------:|
> | HMR | H36M, MI, COCO, LSP, LSPET, MPII | ResNet-50 | ImageNet | 64.55 | 58.20 | 51.8 | 51.66 |
> | SPIN [1] | H36M, MI, COCO, LSP, LSPET, MPII | ResNet-50 | HMR(ImageNet) | 59.00 | 57.08 | 51.54 | 50.69 |
> | GraphCMR [2] | H36M, COCO, LSP, LSPET, MPII, UP3D | ResNet-50 | ImageNet | 70.51 | 67.2 | 61.74 | 60.26 |
> | PARE [3] | H36M, MI, EFT-[COCO, LSPET, MPII] | HRNet-W32 | ImageNet | 61.99 | 61.13 | 59.98 | 58.32 |
> | Graphormer [4] | H36M, MuCo, COCO, UP3D, MPII | HRNet-W48 | ImageNet | 63.18 | 63.47 | 59.66 | 58.82 |
>
>
> > Q2: While the paper claims to "provide strong baselines for fair comparisons of algorithms," baselines are only done for one model (HMR).
>
> A2: Thank you for the comment. We would like to clarify that we intended for HMR to be a base model where we could fix the algorithm so as to evaluate other aspects beyond algorithms, such as the dataset combination, model initialisation and training strategies.  In our revision, we provide some benchmarking results for different algorithms in Table 10. Due to the time constraint of the rebuttal period, we conduct experiments of additional base methods on the optimal configuration suggested by HMR. We will keep working on this, identify the optimal baselines for more algorithms, and update them in our github repository.

---

> > ### Author Response · Authors · 2022-08-18
> > **Response (2/2)**
> >
> > >Q3: The paper does not present a dataset or a benchmark, but rather proposes stronger baselines on existing benchmarks, and recommendations for training strategies - it’s not clear that this is in scope for the DB track.
> >
> > A3: Thanks for the comments. We believe that our paper can fit for the D&B track for the following reasons.
> >
> > First, We refer to the blog post referenced in the FAQs of the D&B track [https://neuripsconf.medium.com/announcing-the-neurips-2021-datasets-and-benchmarks-track-644e27c1e66c] and identified several scopes that are relevant to us:
> > - **“audits of existing datasets, or systematic analysis of existing systems on novel datasets that yield important new insight are also in scope”** – We provide audits of datasets used in prior 3D pose and shape estimation studies and include new ones that have not been used (Table 2). We obtain new insights for what makes a dataset effective for training, which are helpful for future dataset selection, creation or enhancement of existing datasets.
> > - **“As part of this track, we aim to gather advice on best practices in constructing, documenting, and using datasets”** – We provided advice on using and constructing datasets (Remark 1).
> > - **“other benchmarking efforts to connect models to real world impacts”** – We identified optimal configurations that help model training.
> >
> > Second, there is precedence of similar works from NeurIPS’21 D&B track that do not provide new datasets or benchmarking metrics. For instance, similar to our paper, some papers [5, 6, 7] provide analysis on non-data aspects, i.e., architectures or how to build a more robust model; some papers evaluate the benchmarking practices [8, 9]. We believe those papers and ours all fit into this track.
> >
> > Third, model initialization and training strategies are relevant to dataset benchmarking, as the selection of those strategies can highly affect the selection of training datasets, as we discovered in this paper.
> >
> > In data augmentation (section 5.1) under “Training strategies”, we have identified data augmentation benefits indoor datasets more than outdoor datasets. In loss selection (section 5.2) under “Training strategies”, we have identified that L1 loss can be used to tackle noisiness in SMPL labels, a problem specific to 3D pose and shape estimation datasets. In model initialization (Table 7) under the “Backbones” section, we discover that initialization pose estimation datasets is highly effective.
> >
> > References:
> >
> > [1] Kolotouros et al. “Learning to Reconstruct 3D Human Pose and Shape via Model-fitting in the Loop.” ICCV 2019.
> >
> > [2] Kolotouros et al. “Convolutional Mesh Regression for Single-Image Human Shape Reconstruction”. CVPR 2019.
> >
> > [3] Kocabas et al. “PARE: Part Attention Regressor for 3D Human Body Estimation.” ICCV 2021.
> >
> > [4] Lin et al. “Mesh Graphormer”. ICCV 2021.
> >
> > [5 Chen et al. “Benchmarks for Corruption Invariant Person Re-identification” NeurIPS 2021 Datasets and Benchmarks Track.
> >
> > [6] Li et al. "MQBench: Towards Reproducible and Deployable Model Quantization Benchmark." NeurIPS 2021 Datasets and Benchmarks Track.
> >
> > [7] Yi et al. “Benchmarking the Robustness of Spatial-Temporal Models Against Corruptions.” NeurIPS 2021 Datasets and Benchmarks Track.
> >
> > [8 Curth et al. “Really Doing Great at Estimating CATE? A Critical Look at ML Benchmarking Practices in Treatment Effect Estimation”. NeurIPS 2021 Datasets and Benchmarks Track.
> >
> > [9] Bao et al., “It’s COMPASlicated: The Messy Relationship between RAI Datasets and Algorithmic Fairness Benchmarks”.  NeurIPS 2021 Datasets and Benchmarks Track.
> >
> >
> > Please don’t hesitate to let us know if there are any additional clarifications or experiments that we can offer!

---

> ### Author Response · Authors · 2022-08-25
> **Follow-up**
>
> Dear reviewer,
>
> We would like to follow up to check if your concerns have been addressed. In the previous response, we have made the following updates/clarification:
> - Regarding your concern on the choice of HMR (Q1), we have added clarification on our choice of HMR as a base model.
> Following your advice to include more algorithms (Q2), we have added support for our main findings on 4 extra algorithms i.e. SPIN, GraphCMR, PARE, Graphormer (Section 6/ Table 10).
> - Regarding the fit of our paper for the D&B track (Q3), we have identified relevant scopes outlined in the D&B track, added precedence of similar works from NeurIPS’21 D&B track and added support for why model initialization and training strategies are also linked to datasets.
>
> We are happy to answer further questions.

---

### Official Review · Reviewer_3FpX · 2022-07-28
**A comprehensive evaluation of datasets, network models and training strategies for HMR problem**

**Rating:** 6
**Confidence:** 4
**Clarity:** The overall paper is well written and…

**Strengths:**

+ The motivation is clear. I quite agree with the author's view that the current HMR research lacks the analysis of training data sets, and the benchmarking study in this paper is a good attempt to solve this problem.
+ The author has done a lot of research and experiments, and comprehensively analyze the various data sets for training. The tested method covers the mainstream methods, with many interesting and significant recommendations made.
+ The overall paper is well written and easy to follow.

**Weaknesses:**

- The benchmark is still based on previous metrics. The methodology and dataset contribution for the benchmark is limited.
- The coverage of the methods is not thorough, as some network models are missing, like PyMAF [79]. I also understand that it is difficult to incorporate all approaches into an HMR-based framework, and I would like to see the discussion about how to evaluate approaches that differ widely from the HMR-based framework.
- The effectiveness of training methods is closely related to the selected network backbones, and this paper can only test training methods on the most mainstream network models. So I believe the recommendation configuration obtained has certain limitations.

**Additional Feedback:**

As mentioned in the 'Weakness' section, I think the authors should try to compose a more fair data set and corresponding benchmarking metrics based on the analysis. In addition, authors are encouraged to expand the pool of the network models to cover more approaches.

**Correctness:**

I think the claims made in the submission are generally correct. The evaluation methods and experiment design are appropriate while minor concerns are raised in the 'Weaknesses'.

**Documentation:**

The paper doesn't present new datasets. The URL of the project has been provided where codes and analysis data are given. The evaluation method can be clearly reproduced with the released code.

**Ethics:**

No ethical problem is identitified.

**Relation To Prior Work:**

The comprehensive evaluation of data sets and training methods are presented and not involved in previous work. This difference is discussed in the manuscript.

**Summary And Contributions:**

The manuscript presents a comprehensive benchmarking study in the terms of datasets, network backbone and training strategy. The authors analyzed the database characteristics, data enhancement strategy, network structure and loss function design by conducting vast experiments, and improved the accuracy of human body reconstruction with proper training configurations. After testing many aspects, the authors believe that the conducted experiments evaluate the existing methods more fairly, and the recommendation configuration can be used as a new baseline for the human mesh recovery problem.

---

> ### Author Response · Authors · 2022-08-18
> **Response**
>
> We sincerely thank the reviewer for your insightful comments and recognition to this work, especially for acknowledging that many interesting and significant recommendations are made. Below is our detailed response to your questions.
>
> > Q1: The benchmark is still based on previous metrics. The methodology and dataset contribution for the benchmark is limited.
>
> A1: Thanks for the comments. We use the standard metric (PA-MPJPE) and 3DPW Protocol 2 for most experiments on 3D pose and shape estimation tasks. While we did not provide a new metric or methodology, our contributions lie in providing new insights in the aspects of datasets, model initialisation and training strategies, which are helpful for future dataset selection, creation or enhancement of existing datasets. We add a new section to summarize the key findings in Appendix A.
>
>
> > Q2: The coverage of the methods is not thorough, as some network models are missing, like PyMAF [79]. I also understand that it is difficult to incorporate all approaches into an HMR-based framework, and I would like to see the discussion about how to evaluate approaches that differ widely from the HMR-based framework.
>
> A2: Thank you for the suggestion. Following this, we have added new evaluation results about different network models (e.g., SPIN, GraphCMR, PARE, Graphormer) in our revision. Please check Section 6 (Table 10) for more detailed descriptions and analysis. In brief, **our evaluation approaches are general to different algorithms, and the conclusions we obtained from HMR are consistent with other algorithms as well.**
>
> We have included four more baselines (SPIN [1], GraphCMR [2], PARE [3], Graphormer [4]) in the benchmark (Section 6/ Table 10) and have obtained consistent results:
> | Algorithms | Datasets   | Backbone | Initialisation | Normal | L1 | L1+COCO | L1+COCO+Aug |
> |:------:|:-------:|:------:|:-------:|:------:|:------:|:-------:|:------:|
> | HMR | H36M, MI, COCO, LSP, LSPET, MPII | ResNet-50 | ImageNet | 64.55 | 58.20 | 51.8 | 51.66 |
> | SPIN [1] | H36M, MI, COCO, LSP, LSPET, MPII | ResNet-50 | HMR(ImageNet) | 59.00 | 57.08 | 51.54 | 50.69 |
> | GraphCMR [2] | H36M, COCO, LSP, LSPET, MPII, UP3D | ResNet-50 | ImageNet | 70.51 | 67.2 | 61.74 | 60.26 |
> | PARE [3] | H36M, MI, EFT-[COCO, LSPET, MPII] | HRNet-W32 | ImageNet | 61.99 | 61.13 | 59.98 | 58.32 |
> | Graphormer [4] | H36M, MuCo, COCO, UP3D, MPII | HRNet-W48 | ImageNet | 63.18 | 63.47 | 59.66 | 58.82 |
>
> In our implementations, we have already included algorithms similar to HMR (i.e. SPIN, GraphCMR). We regard PyMAF as an improved version of HMR, whereby spatial information is used and PyMAF performs mesh alignment feedback rather than the iterative error feedback adopted by HMR. To demonstrate diversity, we included Mesh Graphormer as a method that differs largely from the HMR-based framework. Mesh Graphormer is a nonparametric approach that regresses vertices instead of parametric coefficients directly from an image.
>
> We will continue to add more algorithms to our open-sourced repository in the future. We think PyMAF would be a valuable addition to validate our findings and we plan to add it in our future works.
>
> > Q3: The effectiveness of training methods is closely related to the selected network backbones, and this paper can only test training methods on the most mainstream network models. So I believe the recommendation configuration obtained has certain limitations.
>
> A3: Thanks for the suggestion. We agree that the training performance is closely related to the network backbones. We respectfully disagree that selecting the mainstream network models for benchmarking has limitations.
>
> First, these models are widely used in 3D human pose estimation tasks, our benchmarking study can provide the optimal baselines built on these models, and they can be readily used by researchers for fair evaluations.
>
> Second, our benchmarking methodology is general. Insights from dataset selection, model initialization and training strategies are not specific to a particular algorithm. Whenever there are new models, we can easily apply our methodology to test these models and identify the optimal configurations. We have included four more baselines (SPIN [1], GraphCMR [2], PARE [3], Graphormer [4]) in the benchmark (Section 6/ Table 10) and have obtained consistent results.
>
> References:
>
> [1] Kolotouros et al. “Learning to Reconstruct 3D Human Pose and Shape via Model-fitting in the Loop.” ICCV 2019.
>
> [2] Kolotouros et al. “Convolutional Mesh Regression for Single-Image Human Shape Reconstruction”. CVPR 2019.
>
> [3] Kocabas et al. “PARE: Part Attention Regressor for 3D Human Body Estimation.” ICCV 2021.
>
> [4] Lin et al. “Mesh Graphormer”. ICCV 2021.
>
> Please don’t hesitate to let us know if there are any additional clarifications or experiments that we can offer!

---

> ### Author Response · Authors · 2022-08-25
> **Follow-up**
>
> Dear reviewer,
>
> We would like to follow up to check if your concerns have been addressed. In the previous response, we have made the following updates/clarification:
> - Regarding your concerns on previous metrics (Q1), we have added clarification of our choice of using PA-MPJPE and 3DPW Protocol 2 as the main metric. In the revised paper, we have also provided additional metrics (MPJPE, PVE and PA-PVE) (Table 1, 2, 4, 6) and benchmarks on 7 other test sets (Table 13).
> - Regarding your concern on the lack of generalisability and limited algorithm (Q2 and Q3), we have added support for our findings on 4 additional algorithms (Section 6/ Table 10).
>
> We are happy to answer further questions.

---

### Author Response · Authors · 2022-08-18
**General Response**

We sincerely thank all the reviewers for your constructive feedback and recognition to this work, especially for acknowledging that this is a solid and inspiring work (Reviewer H6N5), many interesting and significant recommendations are made (Reviewers 3Fpx, SarV), good practices for 3D pose and shape estimation are identified for fairness (Reviewers SarV, 4jjK), systematic studies, detailed analysis and extensive ablation was done (Reviewers gtaS, 4jjk, H6N5). Below, we would like to re-emphasize the benefits of this benchmarking work:

- **Promoting fairness.** Most of the focus on 3D mesh recovery is on algorithm development with less effort on establishing proper evaluation protocols (Reviewer SarV). We demonstrate how commonly under-explored factors like dataset mixing, augmentation strategy and model initialisation can significantly impact reported performance, leading to unfair comparison between different methods. Since each prior work makes different choices in all of the dimensions, it is difficult to make good evidence-based decisions. We believe experiments should be evaluated more fairly and the recommended configuration can be used as a new baseline for human mesh recovery algorithms.
- **Providing strong baselines and recommendations to help in pushing boundaries.** We analysed database characteristics, data enhancement, model initialisation and loss function design used in prior works and on new set-ups, and improved the accuracy of human body reconstruction with proper training configurations. We believe our recommendations on datasets, backbones and training strategies will be helpful for future researchers to build strong models. We provide strong baselines for fair future comparisons. We also aim to optimize a unified protocol, which can be used to standardize the training details and ensure fair comparison of algorithms.

We have significantly revised the paper with new experiments and clarifications to address all the reviewers’ valuable comments. Particularly,

- We have added a new section (Section 6) to benchmark 4 more algorithms i.e. SPIN, GraphCMR, PARE, Graphormer (Table 10) and with optimized configurations (Table 16).
- We have also included more test-set benchmarks i.e. 3DPW, H36M, AGORA validation, MPI-INF-3DHP test, EFT-COCO validation, MuPots3D test, EFT-OCHuman test, EFT-LSPET test (Table 13) and computed the correlation between test-sets (Table 11).
- We have also added additional evaluation metrics (MPJPE, PA-PVE and PVE) to Tables 1, 2, 4, 6.
- We extend Section 7 to include more discussions on recommendations and future works
- We add a new section (Appendix A) to summarize all the findings and open-ended questions in this study.

---

### Meta-Review · Area_Chair_ajWx · 2022-09-11

**Recommendation:** Accept
**Confidence:** 4

**Metareview:**

This paper presents a benchmark (in the sense of a large scale ablation and evaluation of existing works/datasets), considering the effects of dataset, "backbone" and training strategies for the problem of human pose and shape estimation. There is a consensus among reviewers that this paper's main limitation is the (over)emphasis on the human mesh recovery (HMR) framework. The forum discussions provide convincing arguments that despite this, the paper still offers with a major contribution to this albeit limited view of the problem. Another common issue raised by reviewers is whether this paper is suitable for the NeurIPS D&B track. I agree with the authors (and Reviewer BCuk) that this paper's premise is indeed well suited for this track, as described in the call for papers and FAQ. Therefore, I dismiss this as a limitation. Finally, there is a strong outlier review, whose exceptionally low score is obliterating this paper's average score. I have read the discussion in detail and while I find that this reviewer is making interesting points, I do not agree that any of them are serious enough to withhold publication of this work. I am content to see the reviewer's points as providing pointers for other future works that may be more influential than the presented one. The possible existence of superior future studies is not a reason to reject this paper. In conclusion, backed by the strong majority of reviewers, I recommend accepting this paper to the NeurIPS 2022 Datasets and Benchmarks program.

---

### Decision · Program_Chairs · 2022-09-16

Accept